# An RNA sponge controls quorum sensing dynamics and biofilm formation in *Vibrio cholerae*

Michaela Huber[1,6], Anne Lippegaus [1,6], Sahar Melamed [2,3], Malte Siemers [1,4], Benjamin R. Wucher[5], Mona Hoyos [1], Carey Nadell [5], Gisela Storz [2] & Kai Papenfort [1,4] ✉

Small regulatory RNAs (sRNAs) acting in concert with the RNA chaperone Hfq are prevalent in many bacteria and typically act by base-pairing with multiple target transcripts. In the human pathogen *Vibrio cholerae*, sRNAs play roles in various processes including antibiotic tolerance, competence, and quorum sensing (QS). Here, we use RIL-seq (RNA-interaction-by-ligation-and-sequencing) to identify Hfq-interacting sRNAs and their targets in *V. cholerae*. We find hundreds of sRNA-mRNA interactions, as well as RNA duplexes formed between two sRNA regulators. Further analysis of these duplexes identifies an RNA sponge, termed QrrX, that base-pairs with and inactivates the Qrr1-4 sRNAs, which are known to modulate the QS pathway. Transcription of *qrrX* is activated by QrrT, a previously uncharacterized LysR-type transcriptional regulator. Our results indicate that QrrX and QrrT are required for rapid conversion from individual to community behaviours in *V. cholerae*.

Quorum sensing (QS) is the process of cell-cell communication in bacteria. The process involves the production, detection, and response to extracellular signaling molecules called autoinducers. QS allows bacteria to synchronously control processes that are only productive when undertaken in unison by the collective, including various virulence-related functions, such as biofilm formation and toxin production. Thus, QS is a promising target for novel antimicrobial intervention strategies, however, for this concept to become a reality, we must first fully understand model QS systems[1,2].

Marine *Vibrio* species, including the major human pathogen *Vibrio cholerae*, have one of the most thoroughly studied QS systems[3]. In all *Vibrios* studied so far, QS relies on post-transcriptional gene regulation by small regulatory RNAs (sRNAs) called Qrr (quorum regulatory RNA)[4]. While the numbers of Qrr homologs vary among different *Vibrio* species (*V. cholerae* encodes four Qrr homologs, Qrr1-4), all Qrrs act together with the RNA chaperone Hfq to control gene expression

by base-pairing with multiple target mRNAs[5]. Two Qrr-target mRNA interactions are of overarching importance for QS performance in *V. cholerae*. First, Qrr1-4 inhibit the expression of the *hapR* mRNA, which encodes a major regulator of high-cell density behaviors that represses biofilm formation and virulence genes[6]. Second, Qrr2-4 stabilize the mRNA encoding the AphA transcriptional regulator, which antagonizes HapR activity and promotes virulence and biofilm formation[7,8]. Consequently, *V. cholerae* cells deficient for *qrr1-4* expression, or lacking the *hfq* gene, display strongly reduced colonization of mice and fail to produce biofilms[9–12].

In addition to the Qrrs, dozens of Hfq-binding sRNAs have been identified in *V. cholerae*[13] and for few of them a function in QS, virulence, or biofilm formation has been established. For instance, the VqmR sRNA is induced by DPO (3,5-dimethyl-pyrazin-2-ol) autoinducer and inhibits biofilm formation and virulence gene expression[14–16] whereas the VadR sRNA adjusts cell shape and biofilm formation[17].

[1]Friedrich Schiller University Jena, Institute of Microbiology, 07745 Jena, Germany. [2]Division of Molecular and Cellular Biology, Eunice Kennedy Shriver National Institute of Child Health and Human Development, Bethesda, MD 20892-5430, USA. [3]Department of Microbiology and Molecular Genetics, Faculty of Medicine, The Hebrew University of Jerusalem, Jerusalem 9112102, Israel. [4]Microverse Cluster, Friedrich Schiller University Jena, 07743 Jena, Germany. [5]Department of Biological Sciences, Dartmouth College, Hanover, NH 03755, USA. [6]These authors contributed equally: Michaela Huber, Anne Lippegaus. ✉e-mail: kai.papenfort@uni-jena.de

These examples show that Hfq-binding sRNAs are crucial for QS control and collective behavior in *V. cholerae*, however, global studies addressing their regulatory roles in this major pathogen are yet missing.

To close this gap, we employed RIL-seq (RNA-interaction-by-ligation-and-sequencing)[18–20] to identify sRNA-target RNA pairs bound by Hfq in *V. cholerae*. This analysis revealed hundreds of previously unknown sRNA-target interactions at low and high cell densities and led to the discovery of several sponge sRNAs. Sponge sRNAs are a class of non-coding regulators that base-pair with other sRNAs to neutralize their activities[21,22]. Detailed analysis of one sponge sRNA, named QrrX, showed that this regulator specifically binds to the Qrr1-4 sRNAs, which inhibits their regulatory functions. Expression of *qrrX* is controlled by the LysR-type regulator QrrT and together QrrX and QrrT accelerated QS transition in *V. cholerae*. In accordance with this regulatory scheme, lack of *qrrX* facilitates biofilm formation, whereas QrrX over-expression has the opposite effect. Together, our findings reveal the genome-wide impact of Hfq-associated sRNAs on gene expression in *V. cholerae* and identify the QrrX sponge sRNA as a critical regulator of QS-associated collective behaviors.

## Results

### RIL-seq analysis of Hfq in *V. cholerae*

To study Hfq-mediated RNA duplex formation in *V. cholerae*, we performed RIL-seq analysis using *V. cholerae* cells producing a Hfq::3XFLAG protein from the native chromosomal location[13]. Specifically, we collected cells from low and high cell densities (OD$_{600}$ of 0.2 and 2.0, respectively), which we exposed to UV-crosslinking. Next, the Hfq::3XFLAG protein together with its associated RNA ligands was co-immunoprecipitated and RNA ends were trimmed. RNA molecules in close proximity were ligated and prepared for cDNA synthesis and paired-end Illumina sequencing. A *V. cholerae* wild-type strain lacking the 3XFLAG epitope served as a negative control in these experiments. In summary, we found 2889 statistically significant RNA-RNA interaction candidates, supported in total by 847,939 and 493,875 chimeric cDNA reads at low and high cell density, respectively (Fig. 1A, B and Supplementary Data 1). Thus, we conclude that Hfq-mediated RNA-duplex formation is pervasive in *V. cholerae* with thousands of interactions occurring at low and high cell density growth conditions. Of note, RNA duplex formation was independent of the chromosomal locations of the sRNAs and targets (Fig. S1A), showing that sRNAs from

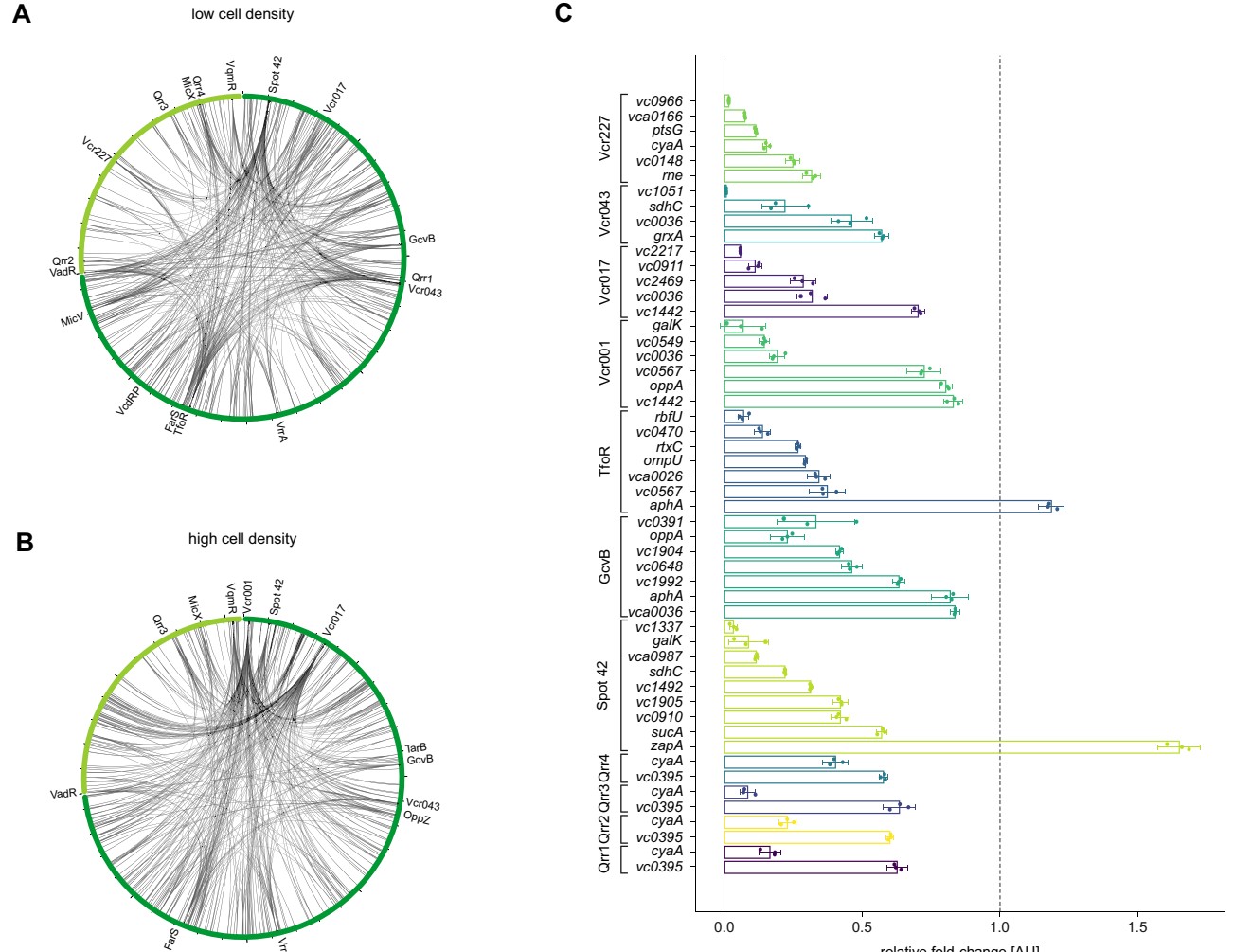

**Fig. 1 | RIL-seq analysis of Hfq in *V. cholerae*. A, B** Circos plots visualizing Hfq-mediated RNA-RNA interactions. *V. cholerae hfq::3XFLAG* cells were cultivated to low (OD600 of 0.2) (**A**) and high cell density (OD600 of 2.0) (**B**) and subjected to RIL-seq analysis. Top 500 significant chimeras are shown and previously reported and sRNAs relevant to **C** are indicated. The first and the second chromosome are marked in dark and light green, respectively. Circos plots were generated using the circos component of the Dash Bio package. **C** Validation of sRNA-mRNA interactions predicted by RIL-seq. Translational GFP reporter fusions were co-transformed with a constitutive sRNA expression plasmid or an empty control plasmid in *E. coli* Top10 cells. GFP production was measured and fluorophore levels from the control strains were set to 1. Bars show mean of independent biological replicates ± propagated SD, *n* = 3. Data belonging to a common regulating sRNA are presented in the same color. Source data underlying panel **C** is provided as a Source Data file.

both *V. cholerae* chromosomes are equally involved in post-transcriptional gene control.

Using a ≥20 chimeric read cut-off, we detected RNA duplex formation for 57 of the 82 previously identified Hfq-binding sRNAs[13] under the tested conditions. Comparison of our dataset with 76 previously described Hfq-dependent sRNA-target mRNA interactions from *V. cholerae* revealed overlap in 35 cases (Table S1). For example, the RIL-seq approach also recovered interactions of Qrr1-4 with *hapR* and Qrr2-4 with *aphA*[4,7]. Using a GFP-based reporter system[23], we further validated post-transcriptional control of 52 target mRNAs by 11 independent sRNAs (Figs. 1C and S1B–I), including well-studied sRNAs such as Qrr1-4, Spot 42, GcvB, and TfoR, as well as the uncharacterized sRNA regulators Vcr001, Vcr017, Vcr043, and Vcr227[13,14]. We chose these sRNAs because they displayed high numbers of chimeric cDNA reads in the RIL-seq experiments (Supplementary Data 1) and engaged interactions with multiple target mRNAs (Figs. S2A–K). Importantly, target validation revealed repressed as well as activated mRNAs indicating that RIL-seq recovers both types of regulation. For example, the Spot 42 sRNA inhibited eight target transcripts, while activating one (Fig. 1C). To promote accessibility of our dataset, we generated a dynamic and searchable web interface that provides a network view of these interactions at http://rnaseqtools.vmguest.uni-jena.de/.

## RIL-seq recovers a high number of sRNA-sRNA interactions
In addition to canonical Hfq-dependent sRNAs affecting gene expression by interacting with *trans*-encoded mRNAs, sRNAs can also base-pair with and inhibit the activity of other sRNAs. These sRNAs are called sponge RNAs[21]. Our RIL-seq dataset identified a total of 81 sRNA-sRNA interactions at both cell densities (Fig. S3). Among these potential sponge RNAs, QrrX (previously identified as Vcr103[13,14]), a yet uncharacterized Hfq-binding sRNA, caught our attention as the data suggested that QrrX base-pairs with all four Qrr sRNAs but no other RNAs (Fig. 2A). The *qrrX* gene is conserved among several *Vibrios*, including pathogenic species such as *Vibrio furnissii*, *Vibrio mimicus*, and *Vibrio anguillarum* (Fig. 2B).

Northern blot analysis of QrrX showed that it is detectable at all stages of growth with peaks of expression when cells transition from low to high cell density ($OD_{600}$ of 1.0) and during late stationary phase (3 h after cells reach an $OD_{600}$ of 2.0) (Figs. 2C and S4A). In contrast, expression of Qrr1-4 peaked at low cell density. In *V. cholerae* cells which were locked at low cell density ($LuxO^{D47E}$)[24] and thus constitutively expressed the Qrr sRNAs, QrrX levels in stationary phase were significantly reduced (Fig. S4B). Conversely, mutation of the *qrrX* gene resulted in elevated Qrr1-4 sRNA levels, while plasmid-borne QrrX production in this background reduced the Qrr1-4 sRNA levels (Fig. S4C). Together, these data indicated that QrrX might act to antagonize Qrr1-4 levels and activity.

## QrrX base-pairs with and destabilizes the Qrr1-4 sRNAs
We next studied the effect of QrrX on Qrr1-4 expression. Given that the QrrX-Qrr1-4 interaction was discovered by RIL-seq (Fig. 2A), we speculated that QrrX would affect Qrr1-4 stability rather than transcription. We tested this hypothesis by pulse induction of QrrX from an inducible plasmid (pBAD-QrrX) for 15 min after which we added rifampicin to halt transcription and followed Qrr1-4 decay by Northern blotting. We chose this experimental setup because transcription of the *qrr1-4* genes is known to be autoregulated[25], which would have complicated the interpretation of the results. In line with previous observations[26], we discovered that the Qrr1-4 sRNAs are highly stable in the absence of QrrX ($t_{1/2} ≥ 32$ min, Fig. 3A, lanes 1–6), whereas over-expression of QrrX led to a drastic reduction in Qrr1-4 stability ($t_{1/2} < 2$ min, lanes 7–12).

Using the *RNAhybrid* algorithm[27], we were able to predict extensive RNA-duplex formation between QrrX and each of the four Qrr sRNAs (Fig. 3B). Interestingly, the base-pairing sequences for both

Qrr1-4 and QrrX are conserved (Figs. 2B and S5A) and for Qrr1-4 the same sequence has previously been identified to participate in base-pairing with *trans*-encoded target mRNAs[4,5]. We tested the predicted base pairing by introducing a single nucleotide exchange (G→C, Fig. 3B) at position 72 of *qrrX* (*qrrX*\* (M1)) and repeated the Qrr1-4 stability experiment. Indeed, this mutation almost fully restored Qrr1-4 stabilities (Fig. 3A, lanes 13–18).

To confirm direct base-pairing of QrrX with the Qrr1-4 sRNAs at the predicted positions, we studied the interaction of QrrX with Qrr1 and Qrr4 in more detail. Specifically, we introduced compensatory single point mutations into the chromosomal genes of *qrr1* and *qrr4* (Fig. 3B) and tested the effect of QrrX or QrrX\* production on the stability of both sRNAs. In both cases, the point mutations rendered Qrr1 and Qrr4 resistant to QrrX over-expression, whereas QrrX\* effectively reduced their stabilities (Fig. 3C, D).

We also tested whether QrrX base-pairing with Qrr1-4 affects QrrX stability and discovered that, while Qrr1-3 reduced the half-life of QrrX (Fig. S5B–D), over-expression of Qrr4 did not have the same effect (Fig. S5E). Despite the almost identical interactions, we noticed that Qrr4 carries a cytosine residue immediately downstream of the conserved base-pairing sequence (nucleotide 49 in Qrr4), whereas Qrr1-3 carry a uridine at the same position (Figs. 3B and S5A). To test the relevance of this residue for Qrr-mediated degradation of QrrX, we changed cytosine to thymidine at the respective position in *qrr4* (*qrr4*\* (M2)) and repeated the experiment. Indeed, over-expression of Qrr4\* (M2) reduced QrrX stability comparable to native Qrr1-3 (Fig. S5F), indicating that base-pairing at the distal end of the interaction dictates the fate of the RNA duplex (Fig. 3B). In summary, we conclude that QrrX base-pairs with all four Qrr sRNAs and that this interaction can affect the stability of the sRNAs.

## RNase E is the major ribonuclease required for turn-over of the QrrX-Qrr4 RNA duplex
To further characterize the fate of the QrrX-Qrr1-4 RNA duplexes, we focused on RNase E (encode by *rne*) and RNase G (encode by *rng*), two partially redundant endoribonucleases with documented roles in sRNA-mediated mRNA degradation[28]. Of note, RNase E is an essential enzyme, however, can be studied in *V. cholerae* using a temperature-sensitive mutant[29]. Using Northern blot analysis, we compared the abundance of QrrX, Qrr1, and Qrr4 in wild-type, Δ*rng*, *rne*^TS, and Δ*rng*/*rne*^TS cells at permissive (30 °C) and non-permissive (44 °C) temperatures. For QrrX we discovered that non-permissive temperatures resulted in slightly increased sRNA abundance (Fig. 4A, lanes 1–2), which was further increased in the absence of RNase G (lanes 3–4) or RNase E (lanes 5–6). Interestingly, absence of both ribonucleases had an additive effect on QrrX levels (lanes 7–8). For Qrr1 and Qrr4, cultivation of *V. cholerae* at non-permissive temperatures resulted in reduced sRNA levels, which were mildly increased in *rne*^TS cells, but remained unaffected in cells lacking RNase G (Fig. 4A).

To determine the contribution of RNase G and RNase E to QrrX-mediated turn-over of the Qrr sRNAs, we determined Qrr4 stability in wild-type, Δ*rng*, *rne*^TS, and Δ*rng*/*rne*^TS cells at non-permissive temperatures. In accordance with our previous results (Fig. 4A), lack of RNase G had only a moderate effect on Qrr4 stability (Fig. 4B; $t_{1/2} = 4$ min vs. 2 min), whereas deactivation of RNase E (either alone or in combination with Δ*rng*) fully restored Qrr4 stability (Figs. 4C and S6; $t_{1/2} > 32$ min). We therefore conclude that RNase E is the key ribonuclease involved in the degradation of the QrrX-Qrr4 RNA duplex.

## Expression of *qrrX* is activated by an uncharacterized LysR-type transcription factor
The *qrrX* gene is located in the intergenic region between the *vca0830* and *vca0832* genes on the second chromosome of *V. cholerae* (Fig. 5A) and this organization is shared among several *Vibrio* species (Fig. S7A).

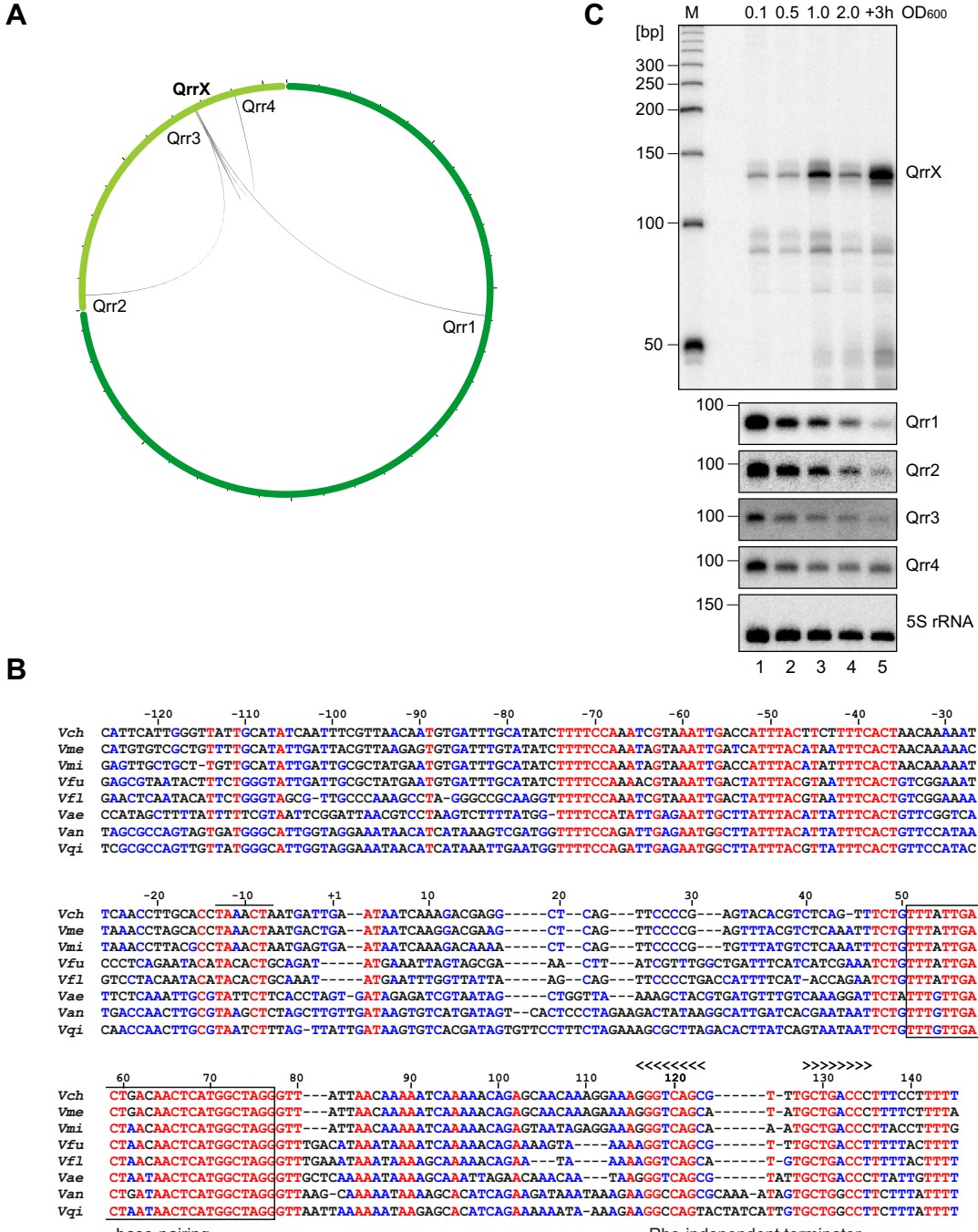

**Fig. 2 | Identification of QrrX sponge RNA. A** Interaction partners of QrrX. Circos plot visualizing interaction partners of QrrX identified by RIL-seq. The first and the second chromosome are marked in dark and light green, respectively. Circos plot was generated using the circos component of the Dash Bio package. **B** Alignment of *qrrX* sequences from various *Vibrio* species. The *qrrX* sequences including the promotor regions were aligned using the Multalin tool[65]. The predicted base-pairing region and the Rho-independent terminator are indicated. Numbers above the sequences indicate the distance to the transcriptional start site. *Vch*, *Vibrio cholerae; Vme, Vibrio metoecus, Vmi, Vibrio mimicus; Vfu, Vibrio furnissii; Vfl, Vibrio fluvialis; Vae, Vibrio aestuarianus; Van, Vibrio anguillarum, Vqi, Vibrio qinghaiensis.* **C** Expression of QrrX. *V. cholerae* wild-type cells were cultivated in LB medium and RNA samples were collected at various stages of growth. Northern blot analysis using specific oligonucleotide probes was performed to determine QrrX, Qrr1, Qrr2, Qrr3, and Qrr4 levels. Probing for 5S ribosomal RNA served as loading control. The experiment was done in three independent biological replicates. Source data underlying panel **C** are provided as a Source Data file.

Conservation analysis indicated that not only the *qrrX* gene is conserved, but also a potential promoter element located between base-pairs 36-74 upstream of the *qrrX* transcriptional start site (Fig. 2B), suggesting a transcriptional regulator recognizes this sequence to regulate *qrrX* expression. To identify this factor, we generated a transcriptional fusion of *qrrX* promoter to *mKate2* and tested fluorescence in *V. cholerae* and *E. coli* cells. We discovered that mKate2 levels were

~11-fold higher in *V. cholerae* cells when compared to *E. coli* (Fig. S7B), indicating that *qrrX* expression is controlled by a *Vibrio*-specific regulator. We harnessed this observation to perform a genetic screen in which we co-transformed a plasmid-based library of the *V. cholerae* genome into *E. coli* cells carrying a *qrrX::lacZ* reporter and screened for blue colonies (Fig. S7C). Among the ~65,000 tested colonies, we isolated 21 candidates displaying a dark blue color and sequencing

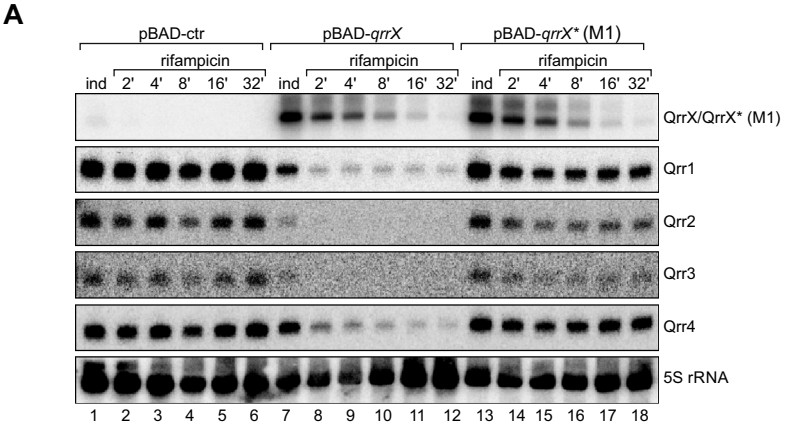

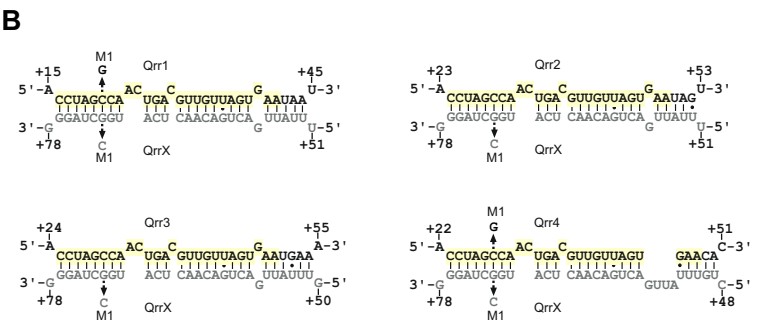

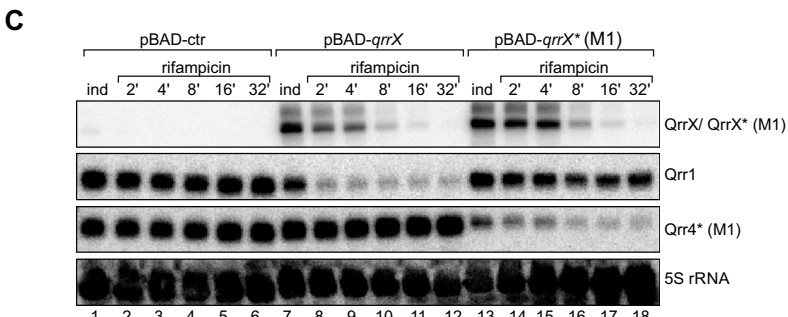

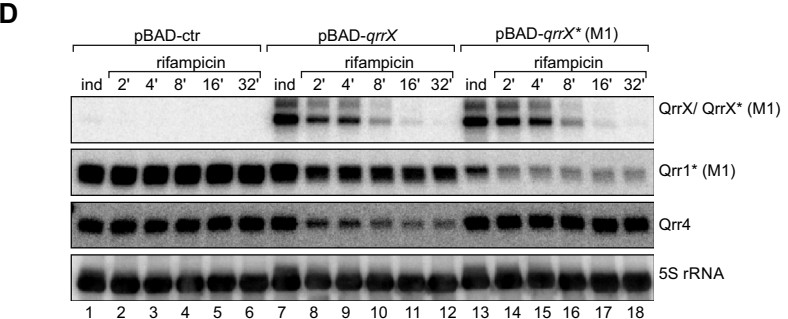

**Fig. 3 | QrrX base-pairs with and destabilizes the Qrr1-4 sRNAs. A** Stability of Qrr1-4 upon induction of QrrX. *V. cholerae* wild-type cells harboring either pBAD-*qrrX*, pBAD-*qrrX*\* (M1) or an empty control plasmid (pBAD-ctr) were cultivated in LB medium to OD600 of 0.2. Expression of QrrX or QrrX\* (M1) was induced with L-arabinose and rifampicin was added to monitor RNA stability. Northern blot analysis shows QrrX/QrrX\* and Qrr1-4 levels at the indicated time points. 5 S ribosomal RNA was used as loading control. The experiment was performed with three independent biological replicates (*n* = 3). **B** Predicted base-pairing interactions between QrrX and Qrr1-4. The sequence that is identical in Qrr1-4 is marked in yellow. Arrows indicate the point mutations in QrrX, Qrr1 and Qrr4 tested in Fig. 3A,

C and D. *RNAhybrid* (Bielefeld BioInformatics Service)[27] was used for prediction. **C, D** Stability of Qrr4\* (M1)/Qrr1\* (M1) upon induction of QrrX. *V. cholerae* cells with a chromosomal point mutation in the *qrr4* gene (M1)/ *qrr1* gene (M1) harboring either pBAD-*qrrX*, pBAD-*qrrX*\* (M1) or an empty control plasmid (pBAD-ctr) were cultivated in LB medium to OD600 of 0.2. Expression of QrrX or QrrX\* was induced with L-arabinose and rifampicin was added to monitor RNA stability. Northern blot analysis shows QrrX/QrrX\*, Qrr1/Qrr1\*, and Qrr4/Qrr4\* levels at the indicated time points. 5 S ribosomal RNA was used as loading control. The experiment was performed with three independent biological replicates (*n* = 3). Source Data underlying panels **A**, **C**, and **D** are provided as a Source Data file.

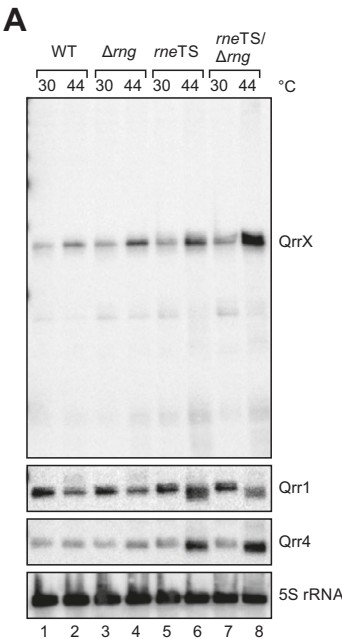

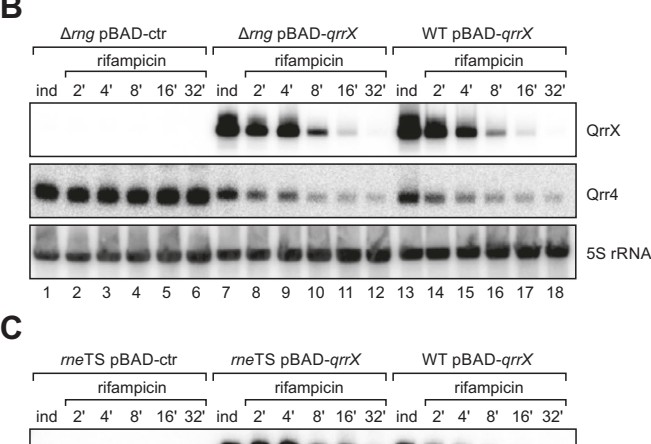

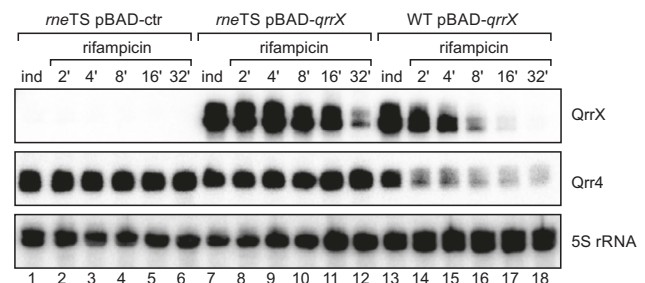

**Fig. 4 | Role of ribonucleases in the QrrX · Qrr1-4 interactions. A** Influence of RNase E and RNase G on QrrX levels. *V. cholerae* wild-type, Δ*rng*, RNase E temperature sensitive (*rne*TS), and Δ*rng rne*TS cells were cultivated at 30 °C to OD600 of 1.0, split in half and either kept at 30 °C or shifted to 44 °C for 60 min. RNA samples were collected and analyzed for QrrX, Qrr1, and Qrr4 levels by Northern blotting. 5 S ribosomal RNA was used as loading control. The experiment was performed with three independent biological replicates (*n* = 3). **B**, **C** Influence of RNase E and RNase G on the stability of the QrrX-Qrr4 duplex. *V. cholerae* wild-type, Δ*rng* (**B**) and *rne*TS (**C**) cells carrying either pBAD-ctr or pBAD-*qrrX* were cultivated at 30 °C to OD600 of 0.2 and then shifted to 44 °C. After 15 min, expression of QrrX was induced with L-arabinose and rifampicin was added to monitor RNA stability. Northern blot analysis shows QrrX and Qrr4 levels at the indicated time points. 5 S ribosomal RNA was used as loading control. The experiment was performed with three independent biological replicates (*n* = 3). Source Data underlying panels **A**–**C** are provided as a Source Data file.

revealed that all 21 candidates contained the gene encoding the LysR transcriptional regulator Vca0830. We validated Vca0830-dependent activation of the *qrrX* promoter in *V. cholerae* using the *qrrX:mKate2* transcriptional reporter (Fig. S7D) and in accordance with the previous nomenclature named this regulator QrrT (QrrX Transcriptional activator). Of note, the *qrrT* gene is located immediately upstream of *qrrX* (Figs. 5A and S7A), suggesting that the two genes are phylogenetically and functionally linked.

To study the role of QrrT on *qrrX* expression, we deleted the *qrrT* gene in *V. cholerae* and performed Northern blot analysis to quantify QrrX and Qrr1-4 levels. When compared to wild-type cells, a *qrrT* mutant had strongly reduced QrrX expression at all stages of growth with residual levels detected in late exponential phase ($OD_{600} = 1.0$) and late stationary phase (3 h after cells reached an $OD_{600}$ of 2.0) (Fig. 5B, lanes 1–10). In accordance with our previous data obtained in Δ*qrrX* cells (Fig. S4C), deletion of *qrrT* also resulted in elevated levels of the Qrr1-4 sRNAs. Plasmid-borne complementation of the Δ*qrrT* mutant restored QrrX and Qrr1-4 levels in *V. cholerae* (Fig. 5B, lanes 11–15), however, this effect was less pronounced in early stationary phase growth conditions ($OD_{600} = 2.0$; lane 14) as we also observed in wild-type cells. We do not think that this phenotype is due to altered turn-over of the QrrX-Qrr1-4 RNA duplex under these conditions given that QrrX stability was highly similar in exponential and stationary phase cells (Fig. S7E). Instead, using Western blot analysis of a chromosomally tagged QrrT::3XFLAG strain, we noticed that QrrT protein levels are constant at all stages of growth (Fig. S7F), indicating that reduced QrrX levels in stationary phase cells might result from changes in the activity of the QrrT transcription factor.

We next compared the activity of *qrrX:mKate2* transcriptional reporter in *V. cholerae* wild-type and Δ*qrrT* cells. Consistent with a direct role of QrrT in *qrrX* transcription activation, *qrrT* deficiency reduced mKate2 production (Fig. 5C; -3.5-fold). We also tested the impact of two conserved *qrrX* promoter elements (sequences P1 and P2, see Fig. S7G) on the performance of this reporter. In both cases, deletion of the respective sequences almost fully abrogated *qrrX:mKate2* activity (Fig. 5C), indicating that these promoter sequences are required for activation by QrrT and potential additional regulators affecting *qrrX* transcription.

We confirmed direct binding of QrrT to the *qrrX* promoter by immunoprecipitation of chromosomally-produced QrrT::3XFLAG protein followed by quantitative PCR of the co-purified DNA (Fig. 5D). When compared to a non-tagged wild-type control, co-immunoprecipitation of QrrT::3XFLAG revealed an -8 fold enrichment of a DNA sequence corresponding to the *qrrX* promoter. In contrast, we observed no enrichment of the *vqmR* promoter sequence, which we used as a negative control in these experiments. Taken together, transcription of *qrrX* is controlled by QrrT, a novel LysR-type transcription factor which binds to and activates the *qrrX* promoter.

## QrrX modulates quorum sensing dynamics
QS is known to have a global impact on gene expression in *V. cholerae* and to modulate several important collective functions[2]. Two transcriptional regulators, called AphA and HapR, are key for many of these functions and both are controlled by the Qrr1-4 sRNAs[7]. Therefore, we asked if by regulating Qrr1-4 sRNAs, QrrX also affects AphA and HapR levels in *V. cholerae*. To this end, we cultivated *V. cholerae* wild-type, Δ*qrrX*, and Δ*qrr1-4* strains from low to high cell densities and monitored AphA and HapR protein levels (using 3XFLAG tagged chromosomal variants) by quantitative Western blot. We discovered that lack of *qrrX* resulted in increased AphA production in cells cultivated to late exponential and stationary phase (Fig. 6A, lanes 1–8). In contrast, HapR levels were decreased in Δ*qrrX* cells under the same growth conditions (Fig. 6B, lanes 1–8). As expected, cells lacking the *qrr1-4* displayed the inverse phenotype showing low AphA and high HapR levels (Fig. 6A, B, lanes 9–12). We were able to confirm differential expression of AphA and HapR in cells lacking *qrrT* or *qrrT* and *qrrX* (Fig. 6C), supporting our

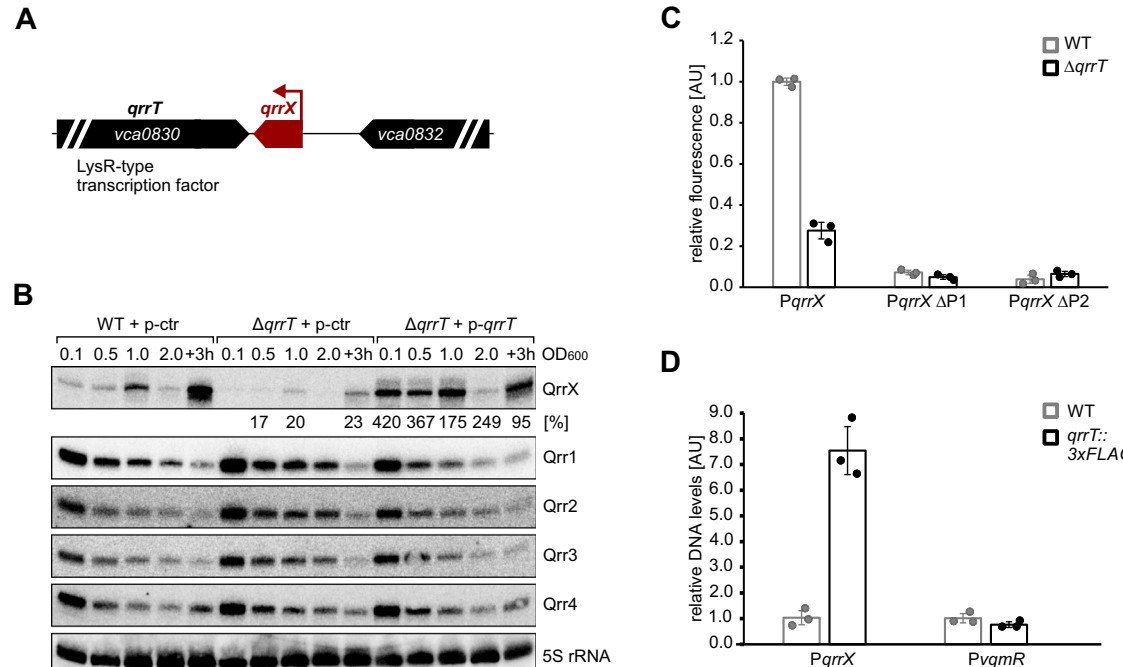

**Fig. 5 | Transcriptional control of *qrrX*. A** Schematic representation of the genomic context of *qrrX*. The *qrrX* gene is marked in red. The *vca0830* gene encodes a LysR-type transcription factor (QrrT). **B** Role of QrrT for QrrX levels. *V. cholerae* wild-type and Δ*qrrT* cells harboring either an empty control vector or a *qrrT* overexpression plasmid (p-*qrrT*) were cultivated in LB medium and RNA samples were collected at different stages of growth. Northern blot analysis was performed to determine QrrX and Qrr1-4 levels. 5 S ribosomal RNA served as loading control. The experiment was performed with three independent biological replicates (*n* = 3). **C** Regulation of the *qrrX* promotor. *V. cholerae* wild-type and Δ*qrrT* cells carrying an mKate2-based transcriptional reporter for *qrrX* (P*qrrX::mKate2*) or a mutated version (P*qrrX::mKate2* ΔP1 or P*qrrX::mKate2* ΔP2, see Fig. S7G) were cultivated in LB medium to OD600 of 1.0 and analyzed for fluorescence. *V. cholerae* wild-type carrying P*qrrX::mKate2* were set to 1. Bars show mean of independent biological replicates ±SD, *n* = 3. **D** ChIP analysis of QrrT. *V. cholerae* wild-type and *qrrT::3XFLAG* cells were cultivated to OD600 of 1.0 and subjected to chromatin immunoprecipitation (ChIP). Bar graphs show relative levels of the *qrrX* and the *vqmR* promotors (P*qrrX* and P*vqmR*), determined by quantitative PCR and P*qrrX* levels in WT *V. cholerae* were set to 1. Data are presented as mean values of independent biological replicates ±SD, *n* = 3. Source Data underlying panels **B**−**D** are provided as a Source Data file.

previous results showing that QrrT activates transcription at the *qrrX* promoter (Fig. 5).

Light production, *i.e.* bioluminescence, is a hallmark of many *Vibrios* and has been instrumental in deciphering the genetic components underlying QS in these organisms[30]. Although *V. cholerae* lacks the ability to produce light, introduction of the *Vibrio harveyi luxCDABE* operon on a plasmid establishes cell density-dependent bioluminescence in *V. cholerae* and can be employed to monitor QS performance[31]. Therefore, to test if QrrX affects QS dynamics in *V. cholerae*, we compared light production of wild-type and Δ*qrrX* cells carrying the *luxCDABE* operon over 10 h of cultivation. We also included the Δ*qrr1-4* mutant in this experiment, which produced high levels of light under all growth conditions[4] and thus served as a positive control. In wild-type and Δ*qrrX* cells, light production decreased immediately after dilution from stationary phase, whereas bioluminescence remained high in cells lacking *qrr1-4* (Fig. 6D). Light production increased sharply with continued growth, yet, wild-type and Δ*qrrX* cells displayed strikingly different bioluminescence kinetics during this transition. Whereas wild-type cells quickly reached maximal light production, lack of *qrrX* resulted in delayed and overall reduced bioluminescence (Fig. 6D). This phenotype is in accordance with the changes in AphA and HapR levels observed in Δ*qrrX* cells (Fig. 6A, B) given that HapR activates the *luxCDABE* operon and that AphA inhibits HapR production[7,31].

### Regulation of biofilm formation by QrrX

In addition to bioluminescence, QS controls various other complex behaviors in *Vibrios*, including biofilm formation[2,32]. In *V. cholerae*, AphA has been reported to induce the synthesis of VpsT, which in turn

activates the production of several structural biofilm components such as the biofilm matrix genes[33,34]. In contrast, HapR inhibits the *vpsT* gene and thus is a negative regulator of biofilm formation[35]. Given that our previous results indicated differential expression of AphA and HapR in cells lacking *qrrX* (Fig. 6A, B), we speculated that QrrX might also affect biofilm formation in *V. cholerae*. To address this question, we measured biofilm formation of wild-type and Δ*qrrX* cells in microfluidic chambers using confocal microscopy. Indeed, after 48 h of incubation, biofilm thickness of the *qrrX* mutant was significantly increased when compared to the wild-type strain (Fig. 6E, F). Conversely, QrrX over-expression inhibited biofilm formation (Fig. S8A, B), whereas over-expression of QrrX* (M1), which is unable to inhibit the Qrr1-4 sRNAs (see Fig. 3A), failed to affect biofilm formation (Fig. S8C). Finally, when seeded at an initial ratio of 1:1, cells lacking the *qrrX* gene had a competitive advantage against isogenic *V. cholerae* wild-type cells, increasing in frequency over 96 h of biofilm growth (Fig. S8D). This result is in accordance with previous results given that AphA activation and HapR repression (as observed in Δ*qrrX* cells) will result in increased biofilm formation and extracellular matrix production[9,33,36,37]. Taken together, our data indicate that QrrX regulates QS-controlled phenotypes, such as bioluminescence and biofilm formation, by base-pairing to and inhibiting the Qrr1-4 sRNAs.

## Discussion

A hallmark of nearly all QS systems is that they allow single-celled organisms to act in unison[2]. This remarkable feature of QS is key to many important collective microbial behaviors, *e.g.* biofilm formation and virulence, and requires tight coordination of signaling events and robust gene regulation control. *Vibrio* species, including *V. cholerae*,

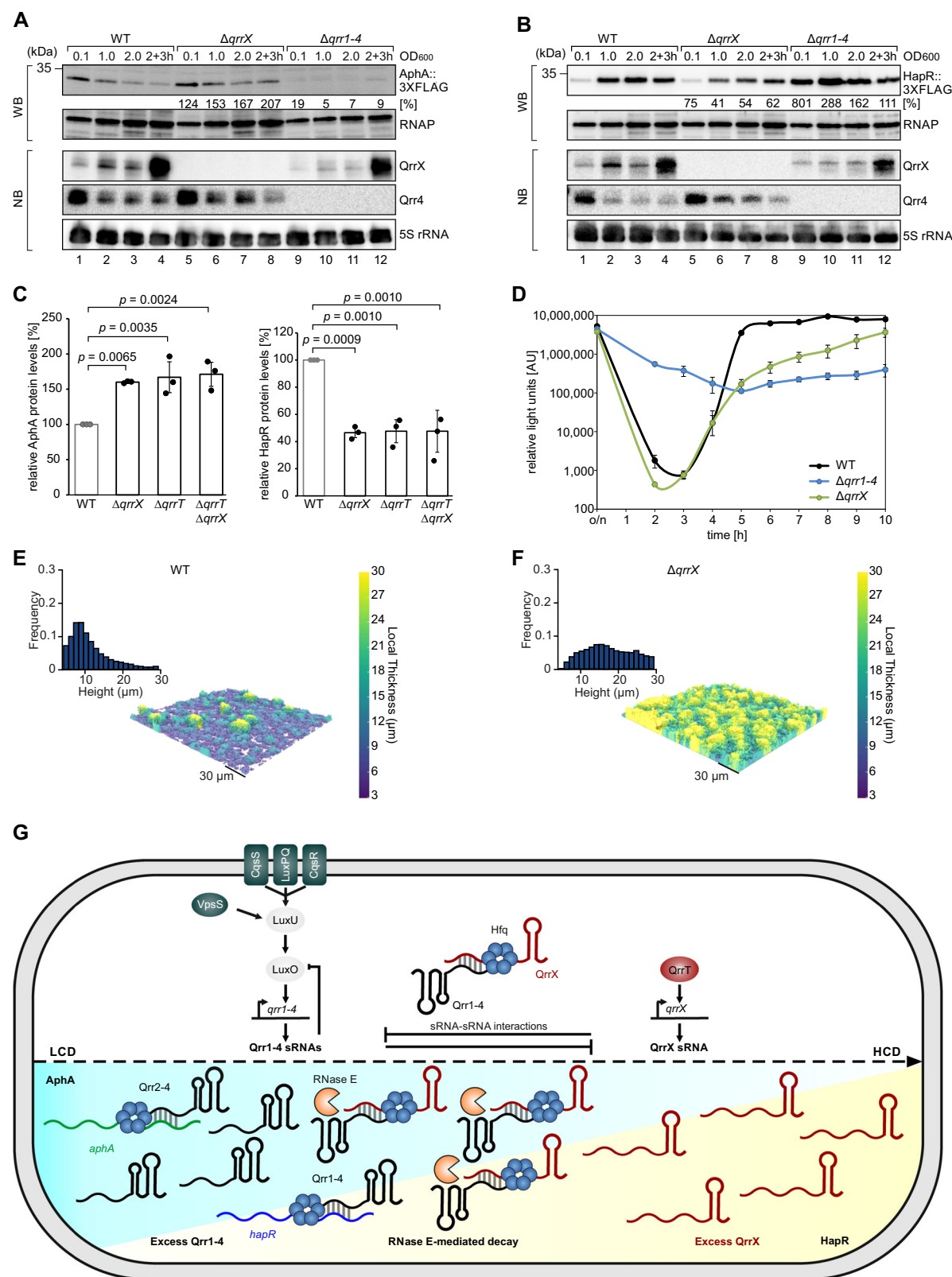

have been intensively studied for their QS architectures providing arguably one of the most thoroughly documented signaling pathways in all bacteria[38]. In this study, we have identified a hitherto unknown component of *V. cholerae*'s QS architecture, the QrrX sponge RNA (Fig. 6G).

Sponge RNAs (a.k.a. decoy RNAs) are RNA molecules that base-pair with and sequester other RNA regulators and have been documented in various biological systems not just limited to Hfq-binding sRNAs[22]. For example, sponge RNAs also occur for sRNAs that act together with the RNA chaperone ProQ[39], as well as RNA-based regulators in eukaryotes, such as microRNAs[40]. However, no sponge RNA has yet been implicated in QS regulation. This might come surprising given that many QS systems rely on sRNAs to achieve optimal performance[41] and that sponge RNAs are effective inhibitors of sRNA

**Fig. 6 | Physiological consequences of QrrX-mediated gene regulation.**
**A**, **B** Influence of QrrX on AphA and HapR. *V. cholerae* wild-type, Δ*qrrX* and Δ*qrr1-4*
cells carrying a chromosomal 3XFLAG-tag at the *aphA* (**A**) or *hapR* (**B**) gene were
cultivated in LB medium, and protein and RNA samples were collected at various
stages of growth. Western blot analysis was performed to monitor AphA (**A**) and
HapR (**B**) protein levels, Northern blot analysis was carried out to determine QrrX
and Qrr4 levels. RNAP and 5S ribosomal RNA served as loading controls for Western
and Northern blots, respectively. The experiment was performed with three inde-
pendent biological replicates (*n* = 3). **C** Influence of QrrX and QrrT on AphA and
HapR. Quantification of AphA-/ HapR-3XFLAG protein levels in *V. cholerae* wild-
type, Δ*qrrX*, Δ*qrrT* and Δ*qrrX*Δ*qrrT* cells carrying a chromosomal 3XFLAG-tag at the
*aphA* or *hapR* gene, respectively. Total protein samples of the indicated strains
were harvested (OD$_{600}$ of 1.0) and tested by Western blot analysis. AphA-/HapR-
3XFLAG protein levels detected in the wild-type cells were set to 100%. Bars show
mean of independent biological replicates ± SD, *n* = 3. Statistical significance was
determined using one-way ANOVA and posthoc Dunnett's multiple comparison
test. Samples of the three biologically independent replicates were processed in
parallel. **D** QrrX modulates QS dynamics. *V. cholerae* wild-type, Δ*qrrX* and Δ*qrr1-4*

cells carrying a QS dependent reporter for bioluminescence were cultivated in SOC
medium, and luminescence was measured at the indicated time points. Error bars
represent SD of three independent biological replicates. **E**, **F** Influence of QrrX on
biofilm accumulation. Biofilms of *V. cholerae* wild-type (**E**) and Δ*qrrX* (**F**) strains
were grown for 48 h and imaged through their whole depth by confocal micro-
scopy. 3D renderings of representative images are color-coded by local thickness.
Bar graphs show frequency distributions of local height for each biofilm.
**G** Regulatory model for QS-mediated gene expression control in *V. cholerae*. Four
QS-specific receptors CqsS, LuxPQ, CqsR, and VpsS respond to external auto-
inducer molecules[66] and modulate the phosphorylation status of the LuxO tran-
scription factor via the LuxU phosphorelay protein. At low cell densities (LCD),
phosphorylated LuxO activates Qrr1-4 expression, which inhibit the *hapR* mRNA,
whereas Qrr2-4 activate AphA expression. In addition, Qrr1-4 also reduce the *luxO*
translation. When transitioning from low to high cell density (HCD), the LysR-type
transcription factor, QrrT, induces the transcription of the QrrX sponge RNA.
Together with Hfq, QrrX binds to and inhibits the Qrr1-4 sRNAs and facilitates
RNase E-mediated decay. Thereby, QrrX accelerates QS dynamics in *V. cholerae*.
Source Data underlying panels **A**–**F** are provided as a Source Data file.

---

activity[21,22]. Indeed, the QrrX sponge RNA solves a long-standing con-
undrum in the *V. cholerae* QS pathway, *i.e.* how does *V. cholerae*
manage to transition rapidly from low to high cell density behavior
despite the relatively long half-life of the Qrr1-4 sRNAs[4,5]? The QrrX
sponge provides a simple, yet elegant answer to this question as it is
able to act specifically on the Qrr1-4 sRNAs by targeting their con-
served base-pairing domain (Fig. 3B). Intriguingly, in the related *V.
harveyi* species, which lack the *qrrX* gene, a different solution to the
same problem has been reported: here, QS dynamics depend on base-
pairing of the Qrr1-5 sRNAs with certain target mRNAs (i.e. *luxM* and
*aphA*) resulting in destabilization of the sRNAs during transition into
high cell density behavior[26]. It is currently unknown why these two
parallel mechanisms exist; however, we speculate that QrrX could
provide a more coordinated QS transition as sponge RNAs typically do
not require translation and our RIL-seq analysis did not identify addi-
tional target transcripts (Fig. 2A). The interaction of Qrr4 with QrrX
might be of special relevance in this process as over-expression of
Qrr1-3 efficiently reduced QrrX stability (Fig. S5B–D), whereas Qrr4 had
only a mild effect on QrrX stability (Fig. S5E). In addition, transcrip-
tional regulation of *qrrX* by QrrT (Fig. 5) provides an additional layer of
control that is missing in *V. harveyi* and related organisms that do not
produce QrrX.

QrrT belongs to the large class of LysR-type transcriptional reg-
ulators, which can be activators or repressors[42]. Structurally, this class
of proteins is characterized by a DNA-binding helix-turn-helix motif at
their N-terminus and a C-terminal co-inducer-binding domain. Tran-
scription control by LysR-type regulators typically involves recogni-
tion of dyadic, often imperfect promoter motifs that guide DNA
binding and indeed we identified two conserved sequence elements in
the *qrrX* promoter and both are necessary for transcriptional activa-
tion (Fig. 5C). DNA-binding is further controlled by co-factor interac-
tion, however, only relatively few co-factors are known and those that
have been identified are chemically highly diverse[42]. We hypothesize
that QrrT activity also relies on co-factor binding given that QrrT
protein levels are constant under all tested conditions (Fig. S7F),
whereas QrrX levels are differentially controlled throughout growth
(Fig. 2C). We currently do not know which molecule(s) interact with
QrrT, however, based on previous transcriptome analyses we can
exclude activation by one of the known *V. cholerae* autoinducer
molecules (*i.e.* AI-2, CAI-1, and DPO)[16]. Future studies aimed at identi-
fying the chemical signal(s) controlling QrrT activity will provide fur-
ther insights into the physiological conditions that govern QS in *V.
cholerae*. Of note, in addition to QrrT, *qrrX* expression might be con-
trolled by other transcriptional regulators, which have not yet been
identified.

In addition to RNA-duplex formation between QrrX and Qrr1-4,
our study revealed hundreds of candidate sRNA-target interactions
(Supplementary Data 1). We have directly validated 52 previously
unknown interactions (Fig. 1C) and predicted base-pairing (Fig. S1B–I).
Whereas additional work will be required to confirm base-pairing at the
indicated positions, several of these interactions suggest interesting
new biology. For example, we discovered that all four Qrr sRNAs
inhibit the *cyaA* mRNA encoding adenylate cyclase (Figs. 1C and S1B),
which catalyzes the synthesis of cyclic AMP (cAMP). cAMP is an
important signaling molecule in nearly all bacteria[43] and in *V. cholerae*
cAMP controls the production of the CAI-1 autoinducer[44]. Thus, our
data indicate the existence of a novel feedback loop that inhibits
premature CAI-1 synthesis by the Qrr1-4 sRNAs. Similarly, our data
suggest that Spot 42 activates the production of ZapA (Figs. 1C
and S1C), a protein which binds to FtsZ and supports cell division[45].
Given that Spot 42 is specifically expressed under high glucose
concentrations[46], one can speculate that activation of *zapA* by Spot 42
facilitates cell replication when nutrients are plentiful. Of note, our
data do not only provide new hypotheses for previously established
sRNA regulators (e.g. Qrr1-4 and Spot 42), but also help to predict the
biological functions of so far uncharacterized regulators. For instance,
three of the four validated targets of the Vcr043 sRNA, i.e. *grxA*,
*vc0036*, and *sdhC* (Figs. 1C and S1H), have documented roles in elec-
tron transport, suggesting that the sRNA could participate in this
process as well. The web interface (http://rnaseqtools.vmguest.uni-
jena.de/) provided together with this work provides a comprehensive
view on all identified interactions including a functional information
about candidate target mRNAs. Thus, the tool offers a quick and effi-
cient means to generate novel hypothesis involving sRNA-mediated
gene control in *V. cholerae* and can further be extended to include
additional organisms for which RIL-seq (or similar) datasets are
available[18–20,39]. In summary, our study offers new opportunities for
hypothesis-driven research approaches focusing on RNA-based gene
expression control in *V. cholerae* including many uncharacterized
sRNAs and regulatory interactions.

## Methods
### Bacterial strains and growth conditions
All strains used in this study are listed in Supplementary Table S3. *V.
cholerae* and *E. coli* strains were cultivated under aerobic conditions in
LB medium at 37 °C, unless stated otherwise. Where appropriate,
antibiotics were used at the following concentrations: 100 µg/ml
ampicillin, 20 µg/ml chloramphenicol, 50 µg/ml kanamycin, 50 U/ml
polymyxin B, 5000 µg/ml streptomycin, 20 µg/ml gentamycin, and
5 µg/ml tetracycline.

## Strain construction

All strains used in this study are listed in Supplementary Table S3. *V. cholerae* C6706 served as wild-type strain. RK2/RP4-based conjugal transfer was used to introduce plasmids form *E. coli* S17λpir donor strains into *V. cholerae*. Subsequently, transconjugants were selected using appropriate antibiotics and polymyxin B to specifically inhibit *E. coli* growth. *V. cholerae* mutant strains were constructed using the pKAS32 suicide vector[47]. Briefly, pKAS32-plasmids were conjugated into *V. cholerae*, and cells were selected for ampicillin resistance. Single colonies were then streaked on fresh plates to select for streptomycin resistance. Desired mutations were confirmed by PCR and sequencing.

## Plasmid construction

All plasmids and all DNA oligonucleotide sequences are listed in Supplementary Tables S2 and S4, respectively. GFP reporter fusions were constructed as previously described[23], and using previously determined transcriptional start sites[23]. The pXG10 vector was used for monocistronic genes, the pXG30 vector for operons[23]. Inserts were amplified from *V. cholerae* genomic DNA with the respective oligonucleotide combinations indicated in the following and cloned into linearized pXG10 (KPO-1702/KPO-1703) via Gibson assembly (GA)[48]: pMH063 (KPO-4210/KPO-4211), pJR026 (KPO-3795/KPO-3796), pJR039 (KPO-4137/KPO-4138), pKT006 (KPO-5411/KPO-5412), pKT001 (KPO-5191/KPO-5192), pMH073 (KPO-4937/KPO-4938), pJR040 (KPO-4132/KPO-4133), pKT008 (KPO-5409/KPO-5369), pKT005 (KPO-5408/KPO-5367), pKT007 (KPO-5410/KPO-5371), pMH067 (KPO-4642/KPO-4643), pMH060 (KPO-4276/KPO-4277), pMD092 (KPO-2573/KPO-2574), pJR029 (KPO-3813/KPO-3814), pMH071 (KPO-4056/KPO-4057), pMH062 (KPO-4212/KPO-4213), pJR044 (KPO-4471/KPO-4472), pJR043 (KPO-4469/KPO-4470), pJR042 (KPO-4184/KPO-4185), pMH059 (KPO-4078/KPO-4079), pMH056 (KPO-4060/KPO-4061), pJR045 (KPO-4473/KPO-4474), pMD161 (KPO-2779/KPO-2780), pJR036 (KPO-4019/KPO-4020), pMH093 (KPO-7574/KPO-7575), and pSM001 (KPO-5372/KPO-5373). For pYH034 (KPO-3005/KPO-3006) and pYH033 (KPO-3003/KPO-3004), pXG10 and respective inserts were digested with NsiI and NheI and ligated. pNP058 (KPO-1708/KPO-1709) and pYH038 (KPO-3054/KPO-3055) were constructed likewise, using pXG10-1C[49]. For pXG30 fusions, backbone was linearized with KPO-4646/KPO-1703, and inserts, amplified with the indicated oligonucleotide combinations, were fused via GA: pKT003 (KPO-5187/KPO-5188), pMH066 (KPO-4651/KPO-4136), pMH072 (KPO-4935/KPO-4936), pKT004 (KPO-5209/KPO-5208). Constitutive sRNA expression plasmids pAL030, pAL032, pAL031, pMH057 and pJR035 were constructed by PCR amplification of the respective sRNA genes from *V. cholerae* genomic DNA using oligonucleotide combinations KPO-7587/KPO-7115, KPO-7603/KPO-7117, KPO-7588/KPO-7119, KPO-4062/KPO-4063, and KPO-3965/KPO-3966, respectively, and cloning via GA into pEVS143[50] vector backbone, linearized with KPO-0092/KPO-1397. Plasmids pMD099, pMH088, pMH092, pMH090, and pMD176 were constructed by amplifying the *qrrX*, *qrr1*, *qrr2*, *qrr3*, and *qrr4* genes from *V. cholerae* genomic DNA using oligonucleotides KPO-2558/KPO-2559, KPO-7114/KPO-7115, KPO-7338/KPO-7117, KPO-7118/KPO7119, and KPO-3779/KPO-3780, respectively, and cloning via GA into pBAD1K (pMD004), linearized with KPO-0196/KPO-1397. Plasmid pMD103 was obtained by site-directed mutagenesis of pMD099 using KPO-3749 and KPO-3750, pAL026 by using pMH088 as template and oligonucleotides KPO-7431 and KPO-7432, and pAL027 by using pMD176 as template and oligonucleotides KPO-7433 and KPO-7434. To construct plasmid pMH086, pMD176 was amplified in two parts with KPO-1529/KPO-7033 and KPO-1525/KPO-7032 and the PCR products were subsequently fused via GA. For pAF012, the promotor region of *qrrX* was amplified from *V. cholerae* genomic DNA with KPO-3676/KPO-3677 and GA was used to insert it into the pCMW-1C-*mKate* vector[16], linearized with KPO-2591/KPO-2592. pMH085 was obtained by PCR amplification of pAF012 in two parts using KPO-1737/KPO-6884 and KPO-1734/KPO-6752 and

reassembly using GA. Site directed mutagenesis of pAF012 using KPO-6753/KPO-6754 yielded in plasmid pMH083. For pKAS32[47] plasmids pJR024, pAF013, pAL033, pMH091, pAL035, pAL036, pASp017, and pMH075, backbone was linearized with KPO-0167 and KPO-0168, and inserts were fused using GA. Up and down flanks were amplified from *V. cholerae* genomic DNA using the following oligonucleotides: KPO-3451/KPO-3452/KPO-3453/KPO-3454 (pJR024), KPO-3741/KPO-3742/KPO-3743/KPO-3744 (pAF013), KPO-3741/KPO-3742/KPO-7604/KPO-7605 (pAL033), KPO-7120/KPO-3750/KPO-7121/KPO-3749 (pMH091), KPO-7122/KPO-7033/KPO-7032/KPO-7123 (pAL035), KPO-7668/KPO-7432/KPO-7431/KPO-7669 (pAL036), and KPO-1872/KPO-1873/KPO-1874/KPO-1875 (pASp017); for pMH075, the insert (*qrrT* coding sequence and 3XFLAG sequence) was generated via IDT gene block synthesis. Plasmid pAS005 was constructed by amplifying the up and down flanks of the *qrrX* gene from genomic DNA with KPO-1301/KPO-1304 and KPO-1302/KPO-1305, and cloning them into pKAS32 via restriction digest with KpnI and AvrII and ligation. To generate pAL001, pMD080 was linearized with oligonucleotides pBAD-ATGrev and pZE-Stop-XbaI, and the *qrrT* gene was amplified from genomic DNA using oligonucleotides KPO-3870/KPO-6321 and subsequently cloned via GA into the linearized vector backbone. For pLH002, the *qrrX* gene including the promotor region was amplified from genomic DNA with oligonucleotides KPO-5419/KPO-5420 and GA was used to insert the PCR product into the pCMW-1[51] vector, linearized with KPO-2592/KPO-2757. To construct plasmid pMD285, the promotor region of *qrrX* was amplified from *V. cholerae* genomic DNA using oligonucleotides KPO-4662/KPO-4663, digested with SpeI and SalI, and ligated into pBBR1-MCS5-*lacZ*[52] backbone, linearized with KPO-4660/KPO-4661. Plasmid pNP012 was generated by amplifying the *qrrX* gene with oligonucleotides KPO-1027/KPO-1027 from genomic DNA and subsequent restriction digest of the insert and the pEVS vector with XbaI and ligation. Plasmid pAF015 was obtained by site-directed mutagenesis of pNP0012 using KPO-3749 and KPO-3750.

## RIL-seq experiments

*V. cholerae* wild-type and *hfq::3XFLAG* strains were cultivated in duplicates in LB medium to low (OD600 of 0.2) and high cell densities (OD600 of 2.0). The experimental part of the RIL-seq protocol was carried out as described by Melamed et al.[53]. Briefly, cells corresponding to 40 OD600 units were subjected to protein-RNA cross-linking, cell lysis and co-immunoprecipitation using anti-FLAG-antibody (Sigma; F1804). Subsequently, the co-immunoprecipitated RNA was treated with RNase A/T1 and T4 RNA ligase. Samples were subjected to proteinase K digestion, and RNA was extracted. RNA was then fragmented and treated with TURBO DNase. Ribosomal RNA was depleted and cDNA libraries were prepared. cDNA libraries were sequenced in paired-end mode on a HiSeq 2500 system (Illumina).

## RIL-seq computational analysis

De-multiplexed raw sequencing reads were checked for quality using fastP[54] and polyX tails, regions of low complexity, as well as low quality tails were removed. The remaining reads were mapped to the *V. cholerae* reference genome (NCBI accession numbers NC_002505.1 and NC_002506.1), using bwa-mem2[55] with default values for the affine gap model. A minimum score of 20 was set to allow for ~4 errors in reads with lengths 36 (read1) and 45 (read2) and the reads were mapped in paired-end mode. Bwa-mem2 handles chimeric reads and produces one alignment per fragment in each read. Alignments, which were assigned to multiple positions were discarded. The alignments resulting from the paired-end mapping were assigned to the annotations of the *V. cholerae* reference genome, including annotations for Vcr001-Vcr107[14] and Vcr200-Vcr230[13] and the predicted 5′UTRs and 3′UTRs, and then grouped and sorted according to the position of the read they originated from. In every such set of collected alignments, each pair of alignments was then classified to be chimeric or not by

checking for both parts to be at least 1000 nt apart from each other on the *V. cholerae* genome, while not sharing the same annotation. Every pair of chimeric alignments was then counted as an interaction between the annotations they belong to. Depending on the number of alignments found in a pair of reads, this procedure can result in a 'single' transcript, or one or more interactions per read pair. Replicates were pooled together, adding each replicates interactions to the pooled sum. After processing all alignments, the resulting interactions were assigned a *p*-value by testing for the significance of the interaction between two annotations against the background of all other interactions using the right tailed Fisher's exact test. The *p*-values were corrected using the method of Benjamini–Hochberg[56]. The interactions were then filtered by their number and their statistical significance using a cut-off of 20 reads per interaction and a false discovery rate of 0.05. This strategy was adapted from previous RIL-seq studies[18,20,39] and allowed us to recover a high number of published sRNA-target mRNA interaction from *V. cholerae* (Table S1).

## Annotation of 5'UTRs and 3'UTRs

To predict transcriptional start and termination sites, we used the data from a dRNA-seq experiment in two conditions ($OD_{600}$ = 0.2 and 2.0, GEO accession GSE62084) and from a previous Term-seq experiment (GEO accession GSE144478)[14,29,57]. Raw reads were checked for quality using fastP[54], and polyX tails, regions of low complexity as well as low quality tails were removed. The remaining reads were mapped to the *V. cholerae* reference genome (NCBI accession numbers NC_002505.1 and NC_002506.1), using bwa-mem2[55] with default values for the affine gap score model. Coverage for every library was computed by counting for every position in the genome, how many alignments were overlapping with it followed by normalization (TPM). The coverage was transformed by computing the difference between every neighboring position and then summing up those differences in a sliding window, assigning the sum to the position with largest difference in the window, thus assigning the height of a plateau above background to the position with steepest increase. The resulting positions and values were checked for their increase above background and absolute height and in case of the dRNA-seq data, a relative increase in the TEX-treated samples. We set a cut-off of 1.3 for the ratio towards background, 3 TPM for absolute value and 1.3 for enrichment in the TEX-treated samples. To use the resulting predictions for transcriptional start and termination sites to define 3'UTRs and 5'UTRs, we first used a heuristic to lower the search space. Between every two genes, the 3'UTR and 5'UTR was set to be at maximum 250 nt long, shortening both uniformly if they were overlapping, not allowing shorter UTRs than 25 nt. In the resulting windows, the positions for start or termination sites were queried respectively and the position corresponding to the largest value was chosen to determine the beginning of a 5'UTR or the end of a 3'UTR.

## Fluorescence measurements

For RIL-seq target validation, GFP fluorescence measurements were performed as described previously[49] with *E. coli* Top10 cells cultivated overnight in LB medium. To measure *qrrX* promotor activity, *V. cholerae* and *E. coli* Top10 cells carrying an mKate2 transcriptional reporter were cultivated in LB medium and samples were collected at the desired growth phase. For all fluorescence measurements, three independent biological replicates were used for each strain. Cells were resuspended in PBS and relative fluorescence was determined using a Spark 10 M plate reader (Tecan). Control samples not expressing fluorescent proteins were used to subtract background fluorescence.

## RNA isolation and Northern blot analysis

Total RNA was extracted and prepared as described previously[58]. For Northern blot analysis, RNA samples were separated on 6% polyacrylamide/7 M urea gels and transferred to Hybond-XL membranes (GE Healthcare). Membranes were hybridized in Roti-Hybri-Quick buffer (Roth) at 42 °C with [$^{32}$P] end-labeled DNA oligonucleotides. Oligonucleotides used for probing are listed in Supplementary Table S4. Membranes were washed in three subsequent steps with SSC (5x, 1x, 0.5x)/0.1% SDS wash buffer. Signals were visualized on a Amersham Typhoon phosphorimager (GE Healthcare) and quantified with GelQuant software (BiochemLabSolutions).

## RNA stability experiments

To monitor RNA stability upon induction of QrrX, Qrr1, Qrr2, Qrr3, or Qrr4 cells were cultivated to the desired growth phase, expression of the respective sRNA was induced with L-arabinose (final concentration: 0.2%) from an arabinose-inducible promotor, and rifampicin (final concentration: 250 µg/ml) was added to stop transcription. RNA samples were collected before and after induction and 2, 4, 8, 16, and 32 minutes after addition of rifampicin. Northern blot analysis with oligonucleotides specific for each sRNA was used to determine RNA levels.

## Genetic screen for transcription factor identification

To identify transcriptional regulators of *qrrX*, *lacZ*-deficient *E. coli* cells harboring a plasmid with the *qrrX* promotor fused to *lacZ* were transformed with a plasmid library expressing random *V. cholerae* genomic fragments[59]. Transformants were plated on LB agar containing 5-Brom-4-chlor-3-indoxyl-ß-D-galactopyranosid (X-gal). ~65,000 colonies were analyzed for ß-galactosidase activity and putative candidates for transcriptional regulators were given for sequencing.

## ChIP and quantitative PCR

*V. cholerae* wild-type and *qrrT::3XFLAG* strains were cultivated to OD600 = 1.0. ChIP experiments were performed as described in *Haycocks* et al.[60], with slight modifications. Briefly, cells were subjected to cross-linking with formaldehyde (1% final conc.) and lysed with FA lysis buffer (50 mM Hepes-KOH pH=7, 150 mM NaCl, 1 mM EDTA, 1% Triton X-100, 0.1% Sodium deoxycholate, 0.1% SDS), containing 4 mg/ml lysozyme. Cross-linked lysates were then subjected to sonication (2×30′ pulses), followed by immunoprecipitation with Protein A Sepharose beads (Sigma, #IP02) and anti-FLAG antibody (Sigma, #F3165) for 90 min. After stringent washing with ChIP wash buffer (10 mM Tris-HCl pH=8, 250 mM LiCl, 1 mM EDTA, 0.5% Nonidet-P40, 0.5% Sodium Deoxylate), samples were eluted in ChIP elution buffer (50 mM Tris-HCl pH=7.5, 10 mM EDTA, 1% SDS) and boiled for 10 min to de-cross-link DNA-protein complexes. DNA was purified by phenol-based extraction and quantitative PCR (qPCR) was performed using GoTaq qPCR Master Mix (Promega, #A6002) and the CFX96 Real-Time PCR System (Bio-Rad). *recA* was used as reference gene. Oligonucleotides used for qPCR are listed in Table S4.

## Western blot analysis

Western blot analysis of FLAG-tagged proteins was carried out as described previously[15]. In brief, samples were separated using SDS-PAGE and subsequently transferred to PVDF membranes. Anti-FLAG antibody (Sigma; F1804) was used for detection. RNAP served as loading control and was detected using anti-RNAP antibody (BioLegend; WP003). Signals were visualized on a Fusion FX EDGE imager and quantified with BIO-1D software (Vilber).

## Bioluminescence assay

*V. cholerae* cells harboring plasmid pBB1[31], which carries the *V. harveyi luxCDABE* operon, were cultivated overnight in SOC broth[61] supplemented with tetracycline, and subsequently diluted 1:1000 into fresh medium. Light production was then measured at the indicated time points during growth of the diluted cultures using a Spark 10 M plate reader (Tecan). Three independent biological replicates were used for each strain.

## Microfluidic assembly

Microfluidic devices were cast in Poly-dimethylsiloxane (PDMS) using soft lithography techniques[62,63]. Each device was bonded to a #1.5 glass coverslip (30 mm width by 60 mm length) via plasma cleaning and heat treatment at 95 °C for 10 min. Each device contained 4 chambers with dimensions 3000 μm x 500 μm x 75 μm (LxWxD). Media was placed into 1 mL BD plastic syringes with 25 gauge needles. These needles were then affixed to #30 Cole Palmer PTFE tubing with an inner diameter of 0.3 mm. This tubing was then placed into holes on the device corresponding to each chamber. The syringes were loaded into a Pico Plus Syringe Pump (Harvard Apparatus). Each device also contains vacuum lines that were installed with an additional piece of PTFE tubing.

## Biofilm assay

Strains were grown overnight at 37 °C shaking in LB medium. Cultures of each strain were then diluted in M9 minimal media with 0.5% glucose and regrown to mid exponential phase ($OD_{600} = 1.0$). Once sufficient optical density was reached, the cultures were inoculated into the chambers of the microfluidic device and left to colonize the surface for 1 h. After this colonization period, a flow rate of 0.1 μL/min was established for the remainder of the experiment. For the competition assay, mid exponential phase cultures of each strain were mixed at 1:1 ratio before inoculation into microfluidic chambers. All biofilm experiments were run at room temperature (24 °C). Individual strain biofilms were imaged at 48 h and competition assays were imaged every 24 h.

## Microscopy and image analysis

Biofilm imaging was done using a Zeiss LSM 880 laser scanning confocal microscope fitted with a 40x/1.2 N.A. water immersion objective. A 488 nm laser line was used to excite sfGFP producing strains and a 594 nm laser was used for the mRuby3 producing strains. Replicate images of each biofilm were taken from independent locations from microfluidic devices inoculated with separate identical cultures. Microscope hardware was run via Zeiss Zen Black software. The 3D confocal image data collected from these replicates was then analyzed using the image analysis framework BiofilmQ[64]. Briefly, confocal image data were processed for segmentation and partitioned into a cubic grid with each cube side approximately 1 cell in length (2μm). Frequency diagrams were generated by using the local thickness parameter. 3D renderings of the biofilms were created using Paraview software utilizing OptiX pathtracer raycasting.

## Reporting summary

Further information on research design is available in the Nature Portfolio Reporting Summary linked to this article.

## Data availability

The demultiplexed sequencing data of the RIL-seq experiments are available at the National Center for Biotechnology Information Gene Expression Omnibus (GEO) under the accession code "GSE198671". Previously published and reanalyzed Term-seq and dRNA-seq sequencing data can be found under the GEO accession codes "GSE144478" and "GSE62084", respectively. Additional raw and analyzed data that support the findings of this study are available from the corresponding author upon request. Source data are provided with this paper.

## Code availability

The code for the RIL-seq analysis is available online at https://github.com/maltesie/ChimericFragments and in this study we used a preliminary version comparable to release "v0.1.0 [https://doi.org/10.5281/zenodo.7326918]" with default configuration.

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

## Acknowledgements

We thank Andreas Starick and Yvonne Greiser for excellent technical support. We thank all members of the Papenfort lab for insightful discussions and suggestions. Work in the Papenfort group was supported by the DFG (PA2820/1–2 and EXC2051 - Project-ID 390713860), the Vallee Foundation, and the European Research Council (StG-758212).

## Author contributions

M.Hu., A.L., S.M., B.R.W., M.Ho., C.N., G.S., and K.P. designed the experiments; M.Hu., A.L., S.M., B.R.W., and M.Ho., performed the experiments; M.Hu., A.L., S.M., M.S., B.R.W., M.Ho., C.N., G.S., and K.P. analyzed data; K.P. and G.S. wrote the manuscript.

## Funding

## Competing interests

The authors declare no competing interests.
