## [Peer Review File · Nature Communications]

An RNA sponge controls quorum sensing dynamics and biofilm formation in *Vibrio cholerae*Reviewer #1 (Remarks to the Author):

Huber et al is a nicely written, well presented manuscript describing the identification of a sponge RNA of the Qrr1-4 sRNAs of *Vibrio cholerae* through the technique of RIL-seq. This has clearly identified many promising targets that will provide many interesting projects in the future. The work is very nicely executed and the story as presented is pretty much complete. Whilst I very much enjoyed reading the manuscript the story is somewhat predictable in the biology that has been discovered in line with other sRNAs and their sponges. However, identifying the sponge RNA that controls the levels of the important sRNAs that are involved in regulating Quorum sensing in *Vibrio* species and it "potentially" completes the regulatory loop at this level (although I am sure there are more surprises). This work is interesting to those studying sRNAs and vibrio biologists. The website is useful for exploring the RILseq data. Below are some comments on the presented data, on the presentation, controls, and interpretation.

Introduction – Line 78 – what is a sponge RNA, if you aren't in the field it needs explanation.

Results

Throughout – rifampicin degradation curves to show half lives are not shown – add to supplementary data.

Figure 2C and S4 – I was left confused by the labelling of the time points and OD. Provide a growth curve image with S4 to explain the different samples that have been taken. What is the authors explanation of the large decrease and then large increase in amount of QrrX from OD 1 to OD 2 (drop) and then back up at +3 hrs? There is discussion of transition between low and high cell density, but this doesn't explain the increase at +3 hrs. Same time points also chosen in 5B.

Levels of Qrr1-4 are clearly different and it must have been previously described which is found in the cell at the highest to lowest level. Quantitation between the levels of Qrr1-4 and QrrX by qRT PCR? Does QrrX have a preference for which Qrr sRNA it interacts with first?

Figure S4B and C – there appears to be discrepancy between levels of Qrr2 and Qrr3 between the 2 figures. B WT Qrr2 is lower and in C WT p-ctr is lower? Also different to fig 3A pBAD ctr lanes.

Figure 3C why showing Qrr1 and then mutation of Qrr4? and S5B Qrr4 and then mutation of Qrr1. This makes it unclear and confusing. Combine into one figure and place in main manuscript.

Fig S5G shows the point mutation M2 of Qrr4 that returns the degradation of QrrX like Qrr1-3. What does QrrX level look like in a Δ qrr1-3 mutant? Might there be some reason that Qrr4 is different? Recycling Qrr4 or QrrX for other targets?

Line 201 and 218 repetition of gene location information.

Line 236 – what growth phase? Why only show one point – would make more sense to show the whole curve and would show the transcriptional control – does it help explain the decrease at OD 2 vs OD 1 and +3 h?

Are there other potential transcriptional units controlled by QrrT?

Figure S5C-G what are the extra bands shown in blots for QrrX?

Is QrrX a processed RNA? Quite a lot of the blots show what looks like a double band, especially in the +3h time points. 2A, 6A, 6B, S4A. Has the ends of the RNA been mapped by RACE? Was there any hint in the dRNA-seq data or the term-seq data? What is happening with the pBAD inducible version, this also looks double?

6A and 6B discrepancy between amounts of QrrX – B WT looks like the previous blots, but A looks a little strange. What is happening in the Δ qrr strain? O.D. 2 is different in both blots.

Minor comments

Line 127 sRNA
Line 245 an ~8 fold
Line 318 come as a surprise

Reviewer #2 (Remarks to the Author):

In their manuscript Huber et al. describe their findings on experimental detection of Hfq-mediated RNA-RNA interactions in *Vibrio cholerae*. For this they performed RIL-seq to assess the Hfq-mediated RNA-RNA interactome globally and then focussed on Quorum Sensing related sRNAs and their targets. There they revealed the RNA sponge QrrX, for which they can show in further experiments that it binds the QS sRNA Qrr1-4, leading to their degradation via an RNaseE-mediated pathway. Furthermore, they show that transcription of QrrX is regulated by QrrT a LysR-like transcriptional regulator.

Overall, the study is well designed and the manuscript written clearly. The conclusions drawn are supported by the data and often additional validation experiments have been performed. In the following you will find some specific comments that should be addressed prior to publication:

Major:

- The RIL-seq experiments seem to be a comparative study between low and high cell densities, as this is explicitly mentioned in the Results (p. 3, ll. 92-93) and the Methods (p. 11, ll 385-386). However, the RIL-seq data is never analysed in a comparative fashion. The authors need to clarify if independent RIL-seq experiments were performed. If not, a reason should be given. If yes, the data should also be analysed comparatively. It would be of major interest to the readers if the RIL-seq method is able to capture differential interaction.
- The authors use a chimeric read cutoff of ≥ 20 , but do not provide the rationale how this value was chosen.
- The description of the "RIL-Seq computational analysis" is difficult to understand and needs to be improved. The basic idea is that the paired-end sequencing generates two reads from the different ends of the cDNA template. If the template was a chimera of two interacting RNAs then the two reads should map/align to two different locations in the genome. In case this is true this is counted as an interaction of the respective annotations.
- Methods, Annotation of 5'UTRs and 3'UTRs: If I understand correct the original data from the mentioned publications was reanalysed. So please provide database accessions to these data sets. Reference 13 belongs to the dRNA-seq data, not the Term-seq.

Minor:

- p.3, ll. 97-100: The reads themselves are not statistically significant, I propose to rephrase the sentence to something like "We found 2889 statistically significant RNA-RNA interaction candidates, supported in total by 847.939 and 493.875 chimeric cDNA reads at low and high cell density, respectively."
- pp.3 & 4, Legend of Figure 2: The cutoff of ≥ 20 chimeric reads is mentioned several times. If I understood correctly this cutoff was applied in general. If this is the case, it should be mentioned once in the Methods section and not again elsewhere.
- Figure 2A: Using a Circos Plot to show the interactions of QrrX is not ideal, because it results in a lot of white space. Maybe you can show an overview of the interacting regions (Fig. 3B) of QrrX with Qrr1-4. This would also show that Qrr1-4 compete for the same region in QrrX.

Reviewer #3 (Remarks to the Author):

Huber et al. explored the targetome of previously known but also uncharacterized Hfq-dependent sRNAs in *Vibrio cholerae* using the RIL-seq approach. They focused on QrrX sRNA and quite convincingly demonstrated that QrrX is acting as an sRNA sponge. Indeed, QrrX directly interacts with the well-characterized Qrr sRNAs and, consequently, modulates quorum sensing,

bioluminescence, and biofilm formation. Finally, they demonstrated that QrrX is under the control of QrrT, an uncharacterized transcription regulator and showed that both are key components of the quorum sensing pathway.

I have no major comments. The manuscript is well written, and the web interface greatly facilitates the exploration of this wealth of information.

Minor comments:

- Lines 64-65/Line 109. According to the mentioned article, all Qrr sRNAs positively regulate *aphA* mRNA.

- Lines 78-79. "Several potential sponge sRNAs" as authors characterized only one of those sRNA-sRNA interactions. Please define "sponge sRNAs". Authors could cite this recent review: 10.1016/j.bbagr.2020.194565 .

- Lines 107-108. "76 previously described Hfq-dependent sRNA-target mRNA interactions from *V. cholerae* revealed overlap in 35 cases." Could authors explain why only ~50% of previously known interactions are recovered using RIL-seq?

- Lines 113-115. "We chose these sRNAs because they displayed high numbers of chimeric cDNA reads in the RIL-seq experiments and engaged interactions with multiple target mRNAs (Figs. S2A-K)" Do authors think that all these interactions are "real"? Please discuss it.

- <http://rnaseqtools.vmguest.uni-120jena.de/> Please check the server availability. I was sometimes not able to access this website.

- Line 128 "it was predicted to base-pair with all four Qrr sRNAs but no other RNAs (Fig. 2A)" Many previously characterized sRNA-mRNA interactions were not validated by the RIL-seq approach. I think that authors cannot be 100% sure that QrrX does not interact with other RNAs. Therefore, it might also be possible that Qrr1-4 sRNAs inhibit the function of QrrX.

- Line 124. "This expression pattern is inversely correlated with the Qrr1-4 sRNAs." Not exactly, especially at OD600 of 1 or 2. Please rephrase this sentence.

- Figure 2C. Numerous bands are visible. Do authors think that QrrX could be processed/cleaved?

- Line 136. "QrrX levels were significantly reduced (Fig. S4B)" Only true at OD600 of 3. Please rephrase this sentence.

- Figure S4C. The effect of *qrrX* deletion is quite convincing. However, the level of Qrr1, Qrr2 and Qrr4 sRNAs does not seem to decrease when *qrrX* is overexpressed. Maybe authors could add densitometric analysis or reformulate their conclusions.

- Line 146. Northern

- Line 158. "this mutation almost fully restored Qrr1-4 stabilities". While the seed sequence is conserved, Qrr2 and Qrr3 seem to be still (partially) degraded in presence of QrrX* (M1). Could authors comment on this discrepancy? This certainly explain why author chose to only focus on Qrr1 and Qrr4 in the following chapters. Consequently, authors should be careful when drawing general conclusions.

- Line 165. Please fuse Figures 3C and S5B.

- Lines 176-177. "QrrX base-pairs with all four Qrr sRNAs and that this interaction results in degradation of the RNA-duplex." This cannot be generalized. Authors showed that the overexpression of "Qrr4 did not have the same effect". The half-life of QrrX seems to be similar in presence or absence of Qrr4 (Figure S5F).

- Figures 3C, 4A and 4C. Please show the 5S profile as control for RNase E inactivation.

- Figure 4C. Why are there two bands in the 5S control in Figure 5C (also in the WT background)?
- Line 202. Just out of curiosity, do authors think that *qrrX* overlaps the 3'UTR of *qrrT*?
- Figure 5B. The deletion of *qrrT* drastically reduces *QrrX* level. However, the expression pattern is similar. Can we imagine that another factor is involved in the temporal expression of *qrrX*?
- Line 242 and Figure 5C. "these promoter sequences are required for activation by *QrrT*". Not only, the deletion of P1 and P2 also abrogate the residual expression in the Δ *qrrT* background.
- Lines 298-299. "cells lacking the *qrrX* gene had a competitive advantage against isogenic *V. cholerae* wild type cells". Please explain why the absence of *QrrX* gives a competitive advantage in tested conditions.
- Lines 330-331. "the sponge RNA itself is not translated and does not interact with other transcripts" Data presented in Figure 2A do not support this assumption.
- Figure 6G. I do not understand why *Qrr1-4* are able to interact with *hapR* in presence of *QrrX* (in excess).

REVIEWER COMMENTS

Reviewer #1 (Remarks to the Author):

Huber et al is a nicely written, well presented manuscript describing the identification of a sponge RNA of the Qrr1-4 sRNAs of *Vibrio cholerae* through the technique of RIL-seq. This has clearly identified many promising targets that will provide many interesting projects in the future. The work is very nicely executed and the story as presented is pretty much complete. Whilst I very much enjoyed reading the manuscript the story is somewhat predictable in the biology that has been discovered in line with other sRNAs and their sponges. However, identifying the sponge RNA that controls the levels of the important sRNAs that are involved in regulating Quorum sensing in *Vibrio* species and it "potentially" completes the regulatory loop at this level (although I am sure there are more surprises). This work is interesting to those studying sRNAs and vibrio biologists. The website is useful for exploring the RILseq data. Below are some comments on the presented data, on the presentation, controls, and interpretation.

We thank the reviewer for her/his overall positive assessment of our work.

1. Introduction – Line 78 – what is a sponge RNA, if you aren't in the field it needs explanation.

We added an additional sentence to the introduction providing requested information.

Results

2. Throughout – rifampicin degradation curves to show half lives are not shown – add to supplementary data.

As requested, we have generated the RNA stability plots. We added this information to the source data (these were too many to include in the supplementary data), which is provided together with this manuscript.

3. Figure 2C and S4 – I was left confused by the labelling of the time points and OD. Provide a growth curve image with S4 to explain the different samples that have been taken. What is the authors explanation of the large decrease and then large increase in amount of QrrX from OD 1 to OD 2 (drop) and then back up at +3 hrs? There is discussion of transition between low and high cell density, but this doesn't explain the increase at +3 hrs. Same time points also chosen in 5B.

Following the reviewer's advice, we added an additional figure to Fig. S4 (new Fig. S4A) to clarify at the which time-point of growth the respective samples were collected. Regarding the fluctuating QrrX levels, we show that QrrX stability is comparable at $OD_{600} = 1$ and $OD_{600} = 2$ (Fig. S7E). We also show that QrrT protein levels remain constant over growth (Fig. S7F). Thus, we suggest that QrrT activity, rather than QrrX stability or QrrT abundance, is causing the change in QrrX abundance during growth. As pointed out in the discussion section, we think that identifying the ligand(s) controlling QrrT activity will be key to addressing this question in more detail (lines 341-347), which we aim to do in a follow-up study.

4. Levels of Qrr1-4 are clearly different and it must have been previously described which is found in the cell at the highest to lowest level. Quantitation between the levels of Qrr1-4 and QrrX by qRT PCR? Does QrrX have a preference for which Qrr sRNA it interacts with first?

We thank the reviewer for this comment. However, we would like to point out that these are Northern blot experiments in which we use radiolabeled oligonucleotides probes that are specific to each of the the Qrr1-4 sRNAs. The main reason for this approach is that the Qrr1-4 sRNAs are very similar in size

and thus cannot be detected as individual transcripts using an oligonucleotide probe targeting a conserved sequence stretch in all four sRNAs. Thus, because we used separate probes with different hybridization energies and labelling efficiencies, these experiments are not suitable for comparing the relative abundances of the four Qrr sRNAs. However, previous RNA-seq experiments (Papenfort et al, 2015, PNAS) indicated that Qrr4 and Qrr3 are the most abundant of the four sRNAs and given the almost identical hybridization energies of the predicted RNA duplexes (Fig. 3B), it seems likely that QrrX frequently interacts with Qrr3 and Qrr4. This view is reflected by the higher number of chimeras we obtained for the QrrX-Qrr3 and QrrX-Qrr4 RNA duplexes, when compared to the chimeras we obtained for QrrX-Qrr2 and QrrX-Qrr1 (see Table S1).

5. Figure S4B and C – there appears to be discrepancy between levels of Qrr2 and Qrr3 between the 2 figures. B WT Qrr2 is lower and in C WT p-ctr is lower? Also different to fig 3A pBAD ctr lanes.

As pointed out in the comment above, the northern blot technology employed here does not allow for the comparison of signal intensities between two (or more) blots.

6. Figure 3C why showing Qrr1 and then mutation of Qrr4? and S5B Qrr4 and then mutation of Qrr1. This makes it unclear and confusing. Combine into one figure and place in main manuscript.

We agree with the reviewer's comment and moved Fig. S5B to the main manuscript (new Fig. 3D).

7. Fig S5G shows the point mutation M2 of Qrr4 that returns the degradation of QrrX like Qrr1-3. What does QrrX level look like in a $\Delta qrr1-3$ mutant? Might there be some reason that Qrr4 is different? Recycling Qrr4 or QrrX for other targets?

We thank the reviewer for this comment. Unfortunately, we think it is extremely difficult to address the question of whether Qrr4 is different. As pointed out in our manuscript, transcription of the Qrr1-4 is controlled by dosage compensation (Svenningsen et al, 2009, EMBO J). Specifically, repression of LuxO synthesis by the Qrr1-4 sRNAs (Tu et al, 2010, Mol Cell) generates a negative feedback loop that calibrates the total levels of Qrr1-4 in the cell. Given that LuxO induces Qrr1-4 transcription, deletion of *qrr1-3* would result in derepression of *luxO*, which in turn will activate Qrr4 transcription. Thus, we do not think that testing QrrX levels in a $\Delta qrr1-3$ will allow us to evaluate the consequence QrrX-Qrr4 duplex formation. Nevertheless, we agree with the reviewer that the interaction of QrrX with Qrr4 might have a role to fine-tune the QrrX levels when cells transition from high to low cell densities. We have addressed this issue in the discussion section of our revised manuscript (lines 328-330).

8. Line 201 and 218 repetition of gene location information.

We modified the sentence in line 218 to avoid this duplication of information.

9. Line 236 – what growth phase? Why only show one point – would make more sense to show the whole curve and would show the transcriptional control – does it help explain the decrease at OD 2 vs OD 1 and +3 h?

As noted in the figure legend, these experiments were performed at OD=1.0. We also analyzed the reporter at different time-points of growth and received similar results (see below). However, given the high stability of the fluorescent reporter protein when compared to the relatively short half-life of QrrX sponge RNA, we do not expect that this experimental setup will allow us to capture the dynamics of QrrX expression that have been detected in the Northern blot experiments.

10. Are there other potential transcriptional units controlled by QrrT?

We thank the reviewer for this comment, however, at this point we cannot answer this question with certainty. In preliminary experiments, we over-expressed QrrT from an inducible promoter for 15 min. and used RNA-seq to determine gene expression changes at a transcriptome level. However, in these experiments, we only discovered up-regulation of QrrX and down-regulation of the Qrr sRNAs. Although these results do not exclude that QrrT regulates additional genes beside *qrrX*, it does not seem to function as a global regulator of gene expression in *V. cholerae* (at least under the conditions we tested). Nevertheless, we aim to explore this question in more detail in a follow-up study.

11. Figure S5C-G what are the extra bands shown in blots for QrrX?

Given that we use a QrrX-specific oligonucleotide probe in these blots, we conclude that these bands are QrrX degradation products. We also noticed that these bands become more pronounced when the Qrr sRNAs are over-expressed, supporting the supposition that base-pairing between QrrX and Qrr1-4 RNAs results in degradation of the RNA duplex. We have marked these bands in Fig.S5C-G accordingly.

12. Is QrrX a processed RNA? Quite a lot of the blots show what looks like a double band, especially in the +3h time points. 2A, 6A, 6B, S4A. Has the ends of the RNA been mapped by RACE? Was there any hint in the dRNA-seq data or the term-seq data? What is happening with the pBAD inducible version, this also looks double?

As pointed out in the comment above, we have evidence supporting the hypothesis that QrrX is processed when it is interacting with Qrr1-4 and that this process involves RNase E (Figs. 4A, 4C, and S6). However, based on dRNA-Seq (Papenfert et al, 2015, PNAS) and Term-Seq data (Hoyos et al, 2020, eLife), we can conclude that full-length QrrX is the most abundant isoform in the cell. As pointed out in more detail in response to comment 16 of reviewer #3, we noticed that a fraction of QrrX transcripts escapes transcription termination at the endogenous promoter and continues transcription into the *qrrT* gene. In case of the plasmid-based pBAD promoter, termination read-through at the *qrrX* terminator is captured at a second terminator sequence (immediately downstream of the *qrrX* terminator) that was included on the plasmid to avoid regulation of plasmid-borne elements (e.g. the antibiotic resistance gene) due to activation of the pBAD promoter.

13. 6A and 6B discrepancy between amounts of QrrX – B WT looks like the previous blots, but A looks a little strange. What is happening in the Δqrr strain? O.D. 2 is different in both blots.

We thank the reviewer for bringing this issue to our attention. We have now repeated the experiments shown in Fig. 6A several times and obtained the expected pattern in QrrX expression. Accordingly, we replaced the previous figure with the new data.

Minor comments

Line 127 sRNA

Corrected

Line 245 an ~8 fold

Corrected

Line 318 come as a surprise

Corrected

Reviewer #2 (Remarks to the Author):

In their manuscript Huber et al. describe their findings on experimental detection of Hfq-mediated RNA-RNA interactions in *Vibrio cholerae*. For this they performed RIL-seq to assess the Hfq-mediated RNA-RNA interactome globally and then focused on Quorum Sensing related sRNAs and their targets. There they revealed the RNA sponge QrrX, for which they can show in further experiments that it binds the QS sRNA Qrr1-4, leading to their degradation via an RNaseE-mediated pathway. Furthermore, they show that transcription of QrrX is regulated by QrrT a LysR-like transcriptional regulator. Overall, the study is well designed and the manuscript written clearly. The conclusions drawn are supported by the data and often additional validation experiments have been performed. In the following you will find some specific comments that should be addressed prior to publication:

We would like to thank the reviewer for the critical assessment of our manuscript. We have addressed the comments below and in our revised manuscript.

Major:

- The RIL-seq experiments seem to be a comparative study between low and high cell densities, as this is explicitly mentioned in the Results (p. 3, ll. 92-93) and the Methods (p. 11, ll 385-386). However, the RIL-seq data is never analysed in a comparative fashion. The authors need to clarify if independent RIL-seq experiments were performed. If not, a reason should be given. If yes, the data should also be analysed comparatively. It would be of major interest to the readers if the RIL-seq method is able to capture differential interaction.

We apologize for the confusion, but the RIL-seq data cannot be analyzed in a comparative fashion. The main reason is that the sRNA and mRNA expression profiles change under different growth conditions and upon changes in Hfq concentration. For example, a high number of Hfq-binding sRNAs are transcribed at high cell density (when compared to low cell density; Papenfort et al, 2015, PNAS), which impacts the overall concentration of sRNAs in the cell and therefore changes the pool of possible sRNA-target mRNA interactions, complicating the analyses. Therefore, while the RIL-seq method provides an excellent snapshot of Hfq-mediated RNA-RNA interactions at a given time-point/condition, it cannot easily be adapted for comparative analyses. Of note, we performed two independent biological replicates for each condition tested, in line with previous publications, e.g. Melamed et al, 2016 Mol Cell).

- The authors use a chimeric read cutoff of ≥ 20 , but do not provide the rationale how this value was chosen.

This number was chosen based on previously published RIL-seq datasets (e.g. Matera et al, 2022, Mol Cell \$\geq 30\$, Melamed et al, 2016 Mol Cell \$\geq 10\$, Melamed et al, 2020, Mol Cell \$\geq 39\$ ) and correlates with a

high number of previously reported sRNA-target interactions that were previously reported for *V. cholerae* (see Table S2). We added this information to the revised manuscript (lines 418-421).

- The description of the "RIL-Seq computational analysis" is difficult to understand and needs to be improved. The basic idea is that the paired-end sequencing generates two reads from the different ends of the cDNA template. If the template was a chimera of two interacting RNAs then the two reads should map/align to two different locations in the genome. In case this is true this is counted as an interaction of the respective annotations.

We revised this section to clarify how the RIL-seq data were analyzed (lines 397-421).

- Methods, Annotation of 5'UTRs and 3'UTRs: If I understand correct the original data from the mentioned publications was reanalysed. So please provide database accessions to these data sets. Reference 13 belongs to the dRNA-seq data, not the Term-seq.

As requested, we added the relevant accession numbers to the manuscript (lines 425-426).

Minor:

- p.3, ll. 97-100: The reads themselves are not statistically significant, I propose to rephrase the sentence to something like "We found 2889 statistically significant RNA-RNA interaction candidates, supported in total by 847.939 and 493.875 chimeric cDNA reads at low and high cell density, respectively."

We thank the reviewer for this comment and rephrased this sentence in our revised manuscript (lines 98-100).

- pp.3 & 4, Legend of Figure 2: The cutoff of ≥ 20 chimeric reads is mentioned several times. If I understood correctly this cutoff was applied in general. If this is the case, it should be mentioned once in the Methods section and not again elsewhere.

We removed this information from the main text.

- Figure 2A: Using a Circos Plot to show the interactions of QrrX is not ideal, because it results in a lot of white space. Maybe you can show an overview of the interacting regions (Fig. 3B) of QrrX with Qrr1-4. This would also show that Qrr1-4 compete for the same region in QrrX.

We thank the reviewer for this comment. While we agree that the Circos Plot in Fig. 2A comes with a relatively large area of white space, we decided to keep this figure since it contrasts well with Fig. 1A, which shows hundreds of interactions. We thus think that this figure nicely illustrates that QrrX specifically interacts with the Qrr1-4 sRNAs. The information that Qrr1-4 compete for the same sequence in QrrX is provided in Fig. 3B.

Reviewer #3 (Remarks to the Author):

Huber et al. explored the targetome of previously known but also uncharacterized Hfq-dependent sRNAs in *Vibrio cholerae* using the RIL-seq approach. They focused on QrrX sRNA and quite convincingly demonstrated that QrrX is acting as an sRNA sponge. Indeed, QrrX directly interacts with the

well-characterized Qrr sRNAs and, consequently, modulates quorum sensing, bioluminescence, and biofilm formation. Finally, they demonstrated that QrrX is under the control of QrrT, an uncharacterized transcription regulator and showed that both are key components of the quorum sensing pathway.

I have no major comments. The manuscript is well written, and the web interface greatly facilitates the exploration of this wealth of information.

We were delighted to hear that the reviewer enjoyed reading our manuscript and that there were no major points of criticism. We have addressed all remaining points in our revised manuscript as detailed below.

Minor comments:

1. - Lines 64-65/Line 109. According to the mentioned article, all Qrr sRNAs positively regulate *aphA* mRNA.

We added an additional reference (Shao et al, 2012, Mol Microbiol) to clarify this issue. Here, the authors showed that Qrr1 lacks a critical sequence element in the 5' end of the sRNA that is required for base-pairing with *aphA*.

2. - Lines 78-79. "Several potential sponge sRNAs" as authors characterized only one of those sRNA-sRNA interactions. Please define "sponge sRNAs". Authors could cite this recent review: 10.1016/j.bbagr.2020.194565 .

We added an additional sentence to the introduction providing the requested information. We also added the indicated reference.

3. - Lines 107-108. "76 previously described Hfq-dependent sRNA-target mRNA interactions from *V. cholerae* revealed overlap in 35 cases." Could authors explain why only ~50% of previously known interactions are recovered using RIL-seq?

We thank the reviewer for this comment. Indeed, we were surprised to see that we obtained only ~50% of the known interactions. However, only a limited number of sRNA-target interactions have been studied in *V. cholerae* to date and thus the previously reported interactions stem from relatively few sRNAs. Specifically, we noticed that interactions involving the VrrA and MicV sRNAs are underrepresented in our dataset. These two sRNAs are induced by the alternative sigma factor E and thus require specific stress conditions (e.g. envelope stress) to be induced. It is quite possible that the VrrA and MicV interactions are underrepresented given that the two time-points used in our RIL-Seq experiments do not involve sigma E induction.

4. - Lines 113-115. "We chose these sRNAs because they displayed high numbers of chimeric cDNA reads in the RIL-seq experiments and engaged interactions with multiple target mRNAs (Figs. S2A-K)" Do authors think that all these interactions are "real"? Please discuss it.

Indeed, we consider these interactions to be "real" given that we were able to validate post-transcriptional regulation using a GFP-based reporter system (Fig. 1C). However, we are aware that the RNA duplexes shown in Figs. 2A-K are based on in silico predictions and thus would need further validation. We have clarified this issue in the discussion section of our revised manuscript (lines 349-353).

- <http://rnaseqtools.vmguest.uni-120jena.de/> Please check the server availability. I was sometimes not able to access this website.

We tested the link and worked fine from all of our computer. However, we noticed that the address indicated above contains a copy-paste error (i.e. the line number of the link in the manuscript), which might have caused the accessibility problems.

5. - Line 128 "it was predicted to base-pair with all four Qrr sRNAs but no other RNAs (Fig. 2A)" Many previously characterized sRNA-mRNA interactions were not validated by the RIL-seq approach. I think that authors cannot be 100% sure that QrrX does not interact with other RNAs. Therefore, it might also be possible that Qrr1-4 sRNAs inhibit the function of QrrX.

We modified this sentence to clarify that this interpretation was only based on the RIL-seq experiments that were presented in this manuscript.

6. - Line 124. "This expression pattern is inversely correlated with the Qrr1-4 sRNAs." Not exactly, especially at OD600 of 1 or 2. Please rephrase this sentence.

As requested, we rephrased this sentence in our revised manuscript (lines 133-135).

7. - Figure 2C. Numerous bands are visible. Do authors think that QrrX could be processed/cleaved?

We do think that QrrX could be processed/cleaved. Please see response to comment #12 of reviewer 1.

8. - Line 136. "QrrX levels were significantly reduced (Fig. S4B)" Only true at OD600 of 3. Please rephrase this sentence.

We thank the reviewer for this comment and revised the sentence accordingly (line 135-136).

9. - Figure S4C. The effect of qrrX deletion is quite convincing. However, the level of Qrr1, Qrr2 and Qrr4 sRNAs does not seem to decrease when qrrX is overexpressed. Maybe authors could add densitometric analysis or reformulate their conclusions.

As described in more detail in response to comment #7 of reviewer 1, this result is not unexpected. Specifically, transcription of the Qrr1-4 sRNAs is subject to dosage compensation control, *i.e.* reduction of the Qrr1-4 levels through post-transcriptional regulation (e.g. by QrrX) will activate the transcription of *qrr1-4* genes (Svenningsen et al, 2009, EMBO J). Thus, to study the effect of QrrX on Qrr1-4 (e.g. Fig. 3A), we employed rifampicin to block transcriptional activation of *qrr1-4* in response to QrrX activity.

10. - Line 146. Northern

Corrected.

11. - Line 158. "this mutation almost fully restored Qrr1-4 stabilities". While the seed sequence is conserved, Qrr2 and Qrr3 seem to be still (partially) degraded in presence of QrrX* (M1). Could authors comment on this discrepancy? This certainly explain why author chose to only focus on Qrr1 and Qrr4 in the following chapters. Consequently, authors should be careful when drawing general conclusions.

As requested by reviewer 1 (see comment #2), we now provide quantification of all stability experiments in the source data of our revised manuscript. Based on these data, we did not find significant differences in the stabilities of Qrr1-4 in the presence of the control plasmid, when compared to the QrrX* (M1) mutation. We chose Qrr1 and Qrr4 for the downstream experiments, because they provided better signals in the Northern blot experiments, when compared to Qrr2 and Qrr3.

12. - Line 165. Please fuse Figures 3C and S5B.

Done, see response to comment #6 of reviewer 1.

13. - Lines 176-177. "QrrX base-pairs with all four Qrr sRNAs and that this interaction results in degradation of the RNA-duplex." This cannot be generalized. Authors showed that the overexpression of "Qrr4 did not have the same effect". The half-life of QrrX seems to be similar in presence or absence of Qrr4 (Figure S5F).

We have modified this sentence to clarify this issue (lines 174-175).

14. - Figures 3C, 4A and 4C. Please show the 5S profile as control for RNase E inactivation.

We assume that the reviewer refers to Figs. 4A, 4C and S6 since these experiments involve the *meTS* strain. We have included the relevant blots in the source data and marked the 9S rRNA signal.

15. - Figure 4C. Why are there two bands in the 5S control in Figure 5C (also in the WT background)?

We apologize for the confusion. This was a technical error that occurred due to slight movement of the blot during the exposure. We have replaced the blot.

16. - Line 202. Just out of curiosity, do authors think that *qrrX* overlaps the 3'UTR of *qrrT*?

Indeed, based on previously obtained dRNA-seq data and Term-Seq data, we find that a small proportion of reads fail to terminate at the *qrrX* terminator and continue transcription into the *qrrT* gene (see screenshot below). However, given that termination read-through only occurs in ~2% for all detected transcripts, we do not think the read-through has a significant effect (at least under the tested conditions).

17. - Figure 5B. The deletion of *qrrT* drastically reduces QrrX level. However, the expression pattern is similar. Can we imagine that another factor is involved in the temporal expression of *qrrX*?

At this point, we cannot exclude that other transcription factors bind to the *qrrX* promoter and activate transcription. However, based on the results from our genetic screen (Fig. S7C) and the data from the Northern blot experiments (Fig. 5B) and transcriptional reporter assays (Fig. 5C), we conclude that QrrT is the most relevant transcription factor under the tested conditions. We address this issue in the discussion section of our revised manuscript (lines 347-349).

18. - Line 242 and Figure 5C. "these promoter sequences are required for activation by QrrT". Not only, the deletion of P1 and P2 also abrogate the residual expression in the Δ qrrT background.

We thank the reviewer for this comment and revised the sentence accordingly (lines 237-239).

19. - Lines 298-299. "cells lacking the *qrrX* gene had a competitive advantage against isogenic *V. cholerae* wild type cells". Please explain why the absence of QrrX gives a competitive advantage in tested conditions.

In the absence of QrrX, repression of the Qrr1-4 sRNA is relieved resulting in increased levels of AphA and reduced HapR production (Figs. 6A and 6B). Given that AphA is an activator biofilm formation through the activation of *vpsT* (Yang et al, 2010, Infect Immun), whereas HapR is a repressor of biofilm formation (Hammer et al, 2003, Mol Microbiol), we conclude that cells lacking *qrrX* display increased biofilm formation and thereby gain a competitive advantage over the isogenic wild-type cells under the tested conditions. We addressed this issue in lines 296-298 of the revised manuscript.

20. - Lines 330-331. "the sponge RNA itself is not translated and does not interact with other transcripts" Data presented in Figure 2A do not support this assumption.

We modified this sentence in the revised manuscript to avoid misinterpretation of our data (lines 326-327).

21. - Figure 6G. I do not understand why Qrr1-4 are able to interact with hapR in presence of QrrX (in excess).

We thank the reviewer for this comment and agree that the position of the Qrr1-4-*hapR* RNA duplex in the figure was confusing. We therefore modified the figure to indicate that Qrr1-4 control *hapR* levels at low cell density rather than high cell density.

Reviewer #1 (Remarks to the Author):

The authors have answered all of my questions and comments satisfactorily (and those of the other reviewers). It is a very nice story and I very much enjoyed reviewing it.

Reviewer #2 (Remarks to the Author):

In their revised version the authors have addressed all my concerns and have substantially improved the manuscript.

Reviewer #3 (Remarks to the Author):

All my concerns have been addressed.